# On the Convergence of Step Decay Step-Size for Stochastic Optimization

**Xiaoyu Wang**
KTH - Royal Institute of Technology
Stockholm, Sweden
`wang10@kth.se`

**Sindri Magnússon**
Stockholm University
Stockholm, Sweden
`sindri.magnusson@dsv.su.se`

**Mikael Johansson**
KTH - Royal Institute of Technology
Stockholm, Sweden
`mikaelj@kth.se`

## Abstract

The convergence of stochastic gradient descent is highly dependent on the step-size, especially on non-convex problems such as neural network training. Step decay step-size schedules (constant and then cut) are widely used in practice because of their excellent convergence and generalization qualities, but their theoretical properties are not yet well understood. We provide convergence results for step decay in the non-convex regime, ensuring that the gradient norm vanishes at an $\mathcal{O}(\ln T/\sqrt{T})$ rate. We also provide near-optimal (and sometimes provably tight) convergence guarantees for general, possibly non-smooth, convex and strongly convex problems. The practical efficiency of the step decay step-size is demonstrated in several large-scale deep neural network training tasks.

## 1 Introduction

We focus on stochastic programming problems on the form

$$\min_{x \in \mathcal{X}} \; f(x) := \mathbb{E}_{\xi \sim \Xi}[f(x; \xi)]. \tag{1}$$

Here, $\xi$ is a random variable drawn from some source distribution $\Xi$ over an arbitrary probability space and $\mathcal{X}$ is a closed, convex subset of $\mathbb{R}^d$. This problem is often encountered in machine learning applications, such as training of deep neural networks. Depending on the specifics of the application, the function $f$ can either be non-convex, convex or strongly convex; it can be smooth or non-smooth; and it may also have additional structure that can be exploited.

Despite the many advances in the field of stochastic programming, the stochastic gradient descent (SGD) method [37, 32] remains important and is arguably still the most popular method for solving (1). The SGD method updates the decision vector $x$ using the following recursion

$$x_{t+1} = \Pi_{\mathcal{X}}(x_t - \eta_t \hat{g}_t) \tag{2}$$

where $\hat{g}_t$ is an unbiased estimate of the gradient (or subgradient) at $x_t$ and $\Pi_{\mathcal{X}}$ is the Euclidean projection onto $\mathcal{X}$. The step-size parameter (learning-rate) $\eta_t > 0$ is critical to control the rate at which the model learns and to guarantee that the SGD iterates converge to an optimizer of (1). Setting the step-size too large will result in iterates which never converge; and setting it too small leads to

35th Conference on Neural Information Processing Systems (NeurIPS 2021)

slow convergence and may even cause the iterates to get stuck at bad local minima. As long as the iterates do not diverge, a large constant step-size promotes fast convergence but only to a (large) neighborhood of the optimal solution. To increase the accuracy, we have to decrease the step-size.

The traditional approach is to decrease the step-size in every iteration, typically as $\eta_0/t$ or $\eta_0/\sqrt{t}$. Both these step-size schedules have been studied extensively and guarantee a non-asymptotic convergence of SGD [31, 26, 36, 19, 38, 15]. However, from a practical perspective, these step-size policies often perform poorly, since they begin to decrease too early. For non-convex problems, such as those which arise in training of deep neural networks, the most successful and popular step-size policy in practice is the step decay step-size [25, 21, 22]. This step-size policy starts with a relatively large constant step-size and then cuts the step-size by a fixed number (called decay factor) at after a given number of epochs. Not only does this step-size result in a faster initial convergence, but it also guarantees that the SGD iterates eventually converge to an exact solution.

The step decay step-size is the default choice in many deep learning libraries, such as TensorFlow [4] and PyTorch [35]; both use a decay rate of $0.1$ and user-defined milestones when the step-size is decreased. Although some recent studies attempt to monitor the optimization process to trigger the milestones when certain conditions are met [27, 50], such approaches are difficult to analyze. Another common approach is to simply divide the targeted number of iterations into $N$ intervals of equal length, and trigger a milestone at the end of each such interval. However, the theoretical properties of this simpler step-size policy is also not yet well understood. To make the problem more manageable theoretically, some related step-size policies that let the length of each segment increase linearly [6] or exponentially [19, 8, 49] are analyzed. On the contrary, we consider the more practical case when the horizon $T$ of the step-decay step-size is divided into several equal parts.

If the number of iterations or horizon $T$ is known apriori, reference [11] analyzes a step decay step-size with a decay rate of $1/2$ applied every $T/\log_2 T$ iterations and establishes a near-optimal $\mathcal{O}(\log_2 T/T)$ convergence rate for vanilla SGD on least-squares problems. However, to our knowledge, there are no convergence guarantees in the literature under more general conditions, *e.g.*, for general strongly-convex, general convex, or non-convex problems. Motivated by this, in this paper, we study the non-asymptotic convergence of SGD with the step decay step-size in more general settings.

## 1.1 Main Contributions

This work establishes novel convergence guarantees for SGD with the step decay step-size on non-convex, convex and strongly convex optimization problems. We make the following contributions:

- We propose a non-uniform probability rule $P_t \propto 1/\eta_t$ for selecting the output in the smooth non-convex setting. Based on this rule, we **(i)** establish a near-optimal $\mathcal{O}(\ln T/\sqrt{T})$ rate for SGD with step decay step-size; **(ii)** improve the results for exponential decay step-size [28]; **(iii)** remove the $\ln T$ factor in the best known convergence rate for the $1/\sqrt{t}$ step-size.
- For strongly convex problems, we establish the following error bounds for the last iterate under step-decay: **(i)** $\mathcal{O}(\ln T/T)$ for smooth problem, which we also prove to be tight; **(ii)** $\mathcal{O}(\ln^2 T/T)$ without the smoothness assumption.
- For the general convex case, we prove that the step decay step-size at the last iterate can achieve a near-optimal convergence rate (up to a $\ln T$ factor).

## 1.2 Related Work

For SGD, the best known bound for the expected error of the $T^{\text{th}}$ iterate is of $\mathcal{O}(1/\sqrt{T})$ when the objective is convex and smooth [32, 12], and of $\mathcal{O}(1/T)$ when the objective is also strongly convex [31, 36]. Without any further assumptions, these rates are known to be optimal. If we restrict our attention to diminishing step-sizes, $\eta_t = \eta_0/t$, the best known error bound for strongly convex and non-smooth problems is of $\mathcal{O}(\ln T/T)$ [38], which is also tight [17]. This rate can be improved to $\mathcal{O}(1/T)$ by averaging strategies [36, 26, 38] or a step decay step-size [19]. For smooth non-convex functions, Reference [12] established an $\mathcal{O}(1/\sqrt{T})$ rate for SGD with a constant step-size $\eta_t = \mathcal{O}(1/\sqrt{T})$. Recently, this error bound has proven to be tight up to a constant, unless additional assumptions are made [9].

Step decay step-sizes were probably first considered for deterministic subgradient methods [14] and [40]. More recently step decay step-size schemes have been employed to improve the convergence rates under various conditions: local growth (convex) [45], Polyak-Lójasiewicz (PL) condition [49], sharp growth (non-convex) [8]. Most of these references consider proximal algorithms (which add a quadratic term $\lambda \|\cdot\|^2$ to the original loss function) and, in addition, perform an extra averaging step between stages. In particular, if the function $f$ is $\mu$-weakly convex (i.e. $f + \frac{\mu}{2} \|\cdot\|^2$ is convex), $\lambda$ is required to be larger than $\frac{\mu}{2}$ [49, 6]. In contrast, we study the performance of step decay step-size for the standard SGD algorithm. Moreover, in [45] the output of each inner-loop is an uniform average. So their complexity reflects the average performance over iterations. But for convex and strongly convex problems, we focus on the final iterate which is the most preferred choice in practice. In fact, most practitioners just use the final iterate.

Reference [6] considers slowly decaying step-sizes, $\sim 1/t$ where $t$ is the stage index. In other words, the step-size is not cut by a constant factor between stages as we do, but it decays (very) slowly. Compared to [19] and [49], where the inner-loop size $S$ grows exponentially, we use a constant value of $S$, which is known to work better in practice. In the extreme case when $S = 1$, step-decay reduces to the exponentially decaying step-size recently studied under the PL condition and a general smoothness assumption [28].

A number of adaptive step-size selection strategies have been proposed for SGD (*e.g.*, [10, 42, 23, 30, 29]), some of which result in step-decay policies [44, 27, 50]. For example, Reference [27] develops a statistical procedure to automatically determine when the SGD iterates with a constant step-size no longer make progress, and then halve the step-size. Empirically, this automatic scheme is competitive with the best hand-tuned step-decay schedules, but no formal guarantees for this observed behaviour are given.

The paper is organized as follows. Notation and basic definitions are introduced in Section 2. In Section 3, we analyze the convergence rates of step decay step-size on nonconvex case and propose a novel non-uniform sampling rule for the algorithm output. The convergence for general convex and strongly convex functions are investigated in Sections A and 4, respectively. Numerical results of our algorithms are presented and discussed in Section 5. Finally, conclusions are made in Section 6.

## 2 Preliminaries

In this part, we will give some definitions and notations used throughout the paper.

**Definition 1.** *(1) The stochastic gradient oracle $\tilde{\mathcal{O}}$ is variance-bounded if for any input vector $\hat{x}$, it returns a random vector $\hat{g}$ such that $\mathbb{E}[\|\hat{g} - \mathbb{E}[\hat{g}]\|^2] \leq V^2$. (2) $\tilde{\mathcal{O}}$ is bounded if for any input vector $\hat{x}$, it returns a random vector $\hat{g}$ such that $\mathbb{E}[\|\hat{g}\|^2] \leq G^2$ for some fixed $G > 0$.*

**Definition 2.** *When the function $f$ is differentiable on $\mathcal{X}$, we say that $f$ is $L$-smooth on $\mathcal{X}$ if there exists a constant $L > 0$ such that $\|\nabla f(x) - \nabla f(y)\| \leq L \|x - y\|$. It also implies that $f(y) \leq f(x) + \langle \nabla f(x), y - x \rangle + \frac{L}{2} \|x - y\|^2$ for any $x, y \in \mathcal{X}$. If $f$ is not differentiable on $\mathcal{X}$, we say that $f$ is $L$-smooth with respect to $x^*$ if $f(x) - f(x^*) \leq \frac{L}{2} \|x - x^*\|^2$, $\forall x \in \mathcal{X}$, with $L > 0$.*

The definition of $L$-smoothness for non-convex functions has been considered by [36].

**Definition 3.** *The function $f$ is $\mu$-strongly convex on $\mathcal{X} \subseteq \mathbb{R}^d$ if $f(y) \geq f(x) + \langle g, y - x \rangle + \frac{\mu}{2} \|y - x\|^2$, $\forall x, y \in \mathcal{X}, g \in \partial f(x)$, with $\mu > 0$.*

**Definition 4.** *The function $f$ is convex on $\mathcal{X}$ if $f(y) \geq f(x) + \langle g, y - x \rangle$ for any $g \in \partial f(x)$ and $x, y \in \mathcal{X}$.*

Throughout the paper, we assume the objective function $f$ is bounded below on $\mathcal{X}$ and let $f^*$ denote its infimum. If $f$ is strongly convex, $x^*$ is the unique minimum point of $f$ and $f^* = f(x^*)$.

**Notation:** Let $[n]$ denote the set of $\{1, 2, \cdots, n\}$ and $\|\cdot\| := \|\cdot\|_2$ without specific mention. We use $\lfloor r \rfloor$ and $\lceil r \rceil$ to denote the nearest integer to the real number $r$ from below and above. For simplicity of notation, we assume that $S$, $N$, $(\log_\alpha T)/2$, $\log_\alpha T$ and $T/\log_\alpha T$ are all integers. The subgradient of the function $f$ on $x$ is denoted by $\partial f(x) := \{v \mid f(y) \geq f(x) + \langle v, y - x \rangle, \forall y\}$.

# 3 Non-asymptotic Convergence for Non-convex Problems

In this section, we provide the first convergence bounds for SGD with step-decay step-sizes on non-convex problems. We also show that our technical approach can be used to improve the best known convergence bounds for both i) standard $1/\sqrt{t}$ step-sizes and ii) exponential decay step-sizes. Before we proceed, we illustrate the theoretical novelty that allows us to derive these results. Most convergence results for SGD on non-convex problems analyze a random iterate drawn from $\{x_t\}_{t=1}^T$ with some probability $P_t$ [12, 13, 28]. For example, reference [12] provides the following result.

**Proposition 3.1.** *Suppose that $f$ is $L$-smooth on $\mathbb{R}^d$ and the stochastic gradient oracle is variance-bounded by $V^2$. If the step-size $\eta_t = \eta_0/\sqrt{t} \leq 1/L$, then*

$$\mathbb{E}[\|\nabla f(\hat{x}_T)\|^2] \leq \frac{f(x_1) - f^*}{\eta_0(\sqrt{T} - 1)} + \frac{LV^2\eta_0(\ln T + 1)}{2(\sqrt{T} - 1)}, \tag{3}$$

*where $\hat{x}_T$ is randomly chosen from $\{x_t\}_{t=1}^T$ with probability $P_t \propto \eta_t$.* [1]

Since $\eta_t$ is decreasing, $P_t \propto \eta_t$ means that initial iterates are given higher weights in the average in Equation (3) than the final iterates. This contradicts the intuition that the iterates become better as the algorithm progresses. Ideally, we should do the opposite, *i.e.* put high weights on the final iterates and low weights on the initial iterates. This is exactly what we do to obtain convergence bounds for step decay step-size: we use the probability $P_t \propto 1/\eta_t$ instead of $P_t \propto \eta_t$ [12, 28] or uniformly sampling [8]. This is especially important when the step-sizes decrease exponentially fast, like in step decay and exponential decay step-sizes. For example, suppose that $\eta_t \propto 0.9^{-t}$ and $T = 100$. Then with $P_t \propto \eta_t$ we pick the output from the first 10 iteration with $65\%$ probability. On the other hand, with $P_t \propto 1/\eta_t$ we pick the output from the last 10 iterations with $65\%$ probability; see Figure 1.

There are other averaging or sampling rules which favor the last iterate in a similar spirit as we do. References [26] and [16] consider SGD with a $1/t$ step-size and analyze the increasing weighted average $\sim t^r$ $(r > 0)$ of the iterates. [6, 46] selects a random iterate as output with increasing probabilities, $\sim t^r$ $(r > 0)$ for stagewise optimization algorithms, but their step-sizes and output probabilities are very different from ours. Especially, for step-decay, their output probabilities are on the form $t^r$ $(r > 0)$, while we use $\alpha^t$ $(\alpha > 1)$ which increases much quicker with $t$, hence our sampling rule favours late iterates to a much higher degree. Moreover, our output distribution is directly related to the step-size, and does not include any new parameters that require hyper-tuning.

In the following three subsections we perform such an analysis to 1) provide convergence bounds for step decay step-size (where no bounds existed before) 2) provide improved convergence rate results for exponential decay step-size, and 3) improve the convergence bound for $1/\sqrt{t}$ step-size.

## 3.1 Convergence Rates under the Step Decay Step-Size

Algorithm 1 describes the SGD algorithm with step decay step-size and our probability rule for selecting the output. Here, and in the analysis below, $N$ is the number of stages, and $\alpha$ is the decay factor by which the step-size is divided at the end of each such stage. In practical deep neural network training, $\alpha = 10$ is the default and $N$ is typically a hyper-parameter selected by experience. For non-convex problems, we will show that it makes sense to let $N$ depend on the desired number of iterations $T$ and the decay factor $\alpha$. Specifically, we will derive results for $N = (\log_\alpha T)/2$ (which is typically a small number) and show that this choice works well in numerical experiments.

**Theorem 3.2.** *Suppose that the non-convex objective function $f$ is $L$-smooth on $\mathbb{R}^d$, the stochastic gradient oracle is variance-bounded by $V^2$, and assume that there exists a constant $\Delta_0 > 0$ such that $\mathbb{E}[f(x_1^t) - f^*] \leq \Delta_0$ for all $t \geq 1$. If we run Algorithm 1 with $T > 1$, $\alpha > 1$, $N \geq 1$, $\eta_0 \leq 1/L$, and $x_1^1 \in \mathbb{R}^d$, then*

$$\mathbb{E}[\|\nabla f(\hat{x}_T)\|^2] \leq \frac{2(\alpha - 1)\Delta_0}{\alpha^2\eta_0} \cdot \frac{N\alpha^N}{T(1 - \alpha^{-N})} + \frac{\eta_0 LV^2(\alpha - 1)}{\alpha^N - 1}. \tag{4}$$

---

[1]In [12], $P_t \propto (2\eta_t - L\eta_t^2)$. If $\eta_t$ is far smaller than $1/L$, we have $(2\eta_t - L\eta_t^2) \approx 2\eta_t$. For simplicity, we rewrite the probability as $P_t \propto \eta_t$ to show the results.

---
**Algorithm 1** SGD with step decay step-size on the non-convex case
---

1: **Input:** initial point $x_1^1$, initial step-size $\eta_0$, decay factor $\alpha > 1$, the number of iterations $T$, outer-loop size $N$
2: **Initialize:** inner-loop size $S = T/N$
3: **for** $t = 1 : N$ **do**
4: $\quad \eta_t = \eta_0/\alpha^{t-1}$
5: $\quad$ **for** $i = 1 : S$ **do**
6: $\quad\quad$ Query a stochastic gradient oracle at $x_i^t$ to get a vector $\hat{g}_i^t$ such that $\mathbb{E}[\hat{g}_i^t] = \nabla f(x_i^t)$
7: $\quad\quad x_{i+1}^t = x_i^t - \eta_t \hat{g}_i^t$
8: $\quad$ **end for**
9: $\quad x_1^{t+1} = x_{S+1}^t$
10: **end for**
11: **Return:** $\hat{x}_T$ is randomly chosen from all the previous iterations $\{x_i^t\}$ with probability $P_i^t = \frac{1/\eta_t}{S\sum_{t=1}^N 1/\eta_t}$ where $i \in [S]$ and $t \in [N]$

---

*Furthermore, if we assume that $N = (\log_\alpha T)/2$, then*

$$\mathbb{E}[\|\nabla f(\hat{x}_T)\|^2] \leq A\frac{\Delta_0}{\eta_0} \cdot \frac{\ln T}{\sqrt{T} - 1} + B\frac{LV^2\eta_0}{\sqrt{T} - 1}, \tag{5}$$

*where $A = (\alpha - 1)/(\alpha^2 \ln \alpha)$ and $B = \alpha - 1$.*

The theorem establishes the first convergence guarantee of SGD with step decay step-sizes on non-convex problems. The results in Eq. (4) capture the influence of $N$ (the number of stages) on the convergence rate. For example, if $N = 1$, then the step-decay schedule reduces to the constant step-size and the convergence bound in Theorem 3.2 matches the known results by setting $\eta_t = \eta_0/\sqrt{T}$ [12]. For $N > 1$, the error bound becomes more complicated. Increasing $N$ reduces the noise term, but it increases the error related to the function value. So there is a trade-off between making the two terms in the right hand side of Eq. (4) small.

The convergence rate is optimized by selecting $N = (\log_\alpha T)/2$, which yields the rate given in Eq. (5). It ensures a $\mathcal{O}(\ln T/\sqrt{T})$ rate towards a stationary point, which is comparable to the results for $\eta_t = \mathcal{O}(1/\sqrt{t})$ step-size in [12] or Proposition 3.1. However, as illustrated in our experiments in Section 5, step decay step-size converges faster in practice and tends to find stationary points that generalize better. Moreover, in the deterministic case when $V = 0$, let $N = 1$, Theorem 3.2 yields the standard $\mathcal{O}(1/T)$ convergence rate result for deterministic gradient descent [33, 5, 13]. Another benefit of Theorem 3.2 is that we can reduce the effect of the very large noise variance $V^2$ to the convergence bound by setting $\alpha = 1 + 1/V^2$. This is in contrast to Proposition 3.1 and $1/\sqrt{t}$ step-sizes, where an increased $V^2$ results in a worse upper bound.

The main novel difficulty in proofing Theorem 3.2, compared to similar results, appears in that we have the new term $\sum_{t=1}^T \mathbb{E}[f(x_t) - f(x_{t+1})]/\eta_t^2$ to account for. To this end, we require that the expectation of the function value at each outer iterate $\mathbb{E}[f(x_t)]$ is uniformly upper bounded. In [39], the authors proved that the function values at the iterates of SGD can be controlled by the initial state, provided that the step-size is bounded by $1/L$. Therefore, it is fair to make this assumption as long as the initial state is given. Nevertheless, this assumption (or its stronger version that the objective function is upper bounded) has been widely used (or implied) in optimization [20, 47, 48, 46] and statistic machine learning [43, 7], and it has never been violated in our numerical experiments. We illustrate the details of the proof in the supplementary material.

## 3.2 A Special Case: Exponentially Decaying Step-Size

An interesting special case occurs when we set $S = 1$ in the step decay step-size. In this case, the step-size will decay exponentially, $\eta_t = \eta_0/\alpha^t$. The first convergence rate results for such step sizes, which we will call Exp-Decay, have only recently appeared in the preprint [28]; we compare our work to those results below.

Clearly, for $S = 1$, the exponential decay factor $\alpha > 1$ cannot be chosen arbitrarily: if $\alpha$ is too large, then $\eta_t$ will vanish in only a few iterations. To avoid this, we choose a similar form of the decaying

factor as [28] and set $\alpha = (\beta/T)^{-1/T}$, where $\beta \in [1, T)$. The intuition is that at the final iteration $T$ the step-size is on the order of $1/\sqrt{T}$ (setting $\beta = \mathcal{O}(\sqrt{T})$), and thus does not vanish over the $T$ iterations. Note that in the special case where $S = 1$, then $N = T$, the choice for $\beta$ is also implied from Theorem 3.2 by setting $\alpha^T = \sqrt{T}$. We are now ready to prove the algorithm's convergence.

**Theorem 3.3.** *Suppose that the non-convex objective function $f$ is $L$-smooth and there exists a constant $\Delta_0 > 0$ such that $\mathbb{E}[f(x_t) - f^*] \leq \Delta_0$ for all $t \geq 1$, and that the stochastic gradient oracle is variance-bounded by $V^2$. If we run Algorithm 1 with $T > 1$, $S = 1$, $\eta_0 \leq 1/L$, $x_1^1 \in \mathbb{R}^d$, and $\alpha = (\beta/T)^{-1/T}$, then*

$$\mathbb{E}[\|\nabla f(\hat{x}_T)\|^2] \leq \frac{\eta_0 \ln(\frac{T}{\beta})}{(\frac{T}{\beta} - 1)T} \left[ \frac{2\Delta_0}{\eta_0^2} \left(\frac{T}{\beta}\right)^2 + LV^2 T \right],$$

*where $\hat{x}_T$ is randomly drawn from $\{x_t\}_{t=1}^T$ with probability $P_t = \frac{1/\eta_t}{\sum_{t=1}^T 1/\eta_t}$. In particular, $\beta = \sqrt{T}$ yields*

$$\mathbb{E}[\|\nabla f(\hat{x}_T)\|^2] \leq \left(\frac{\Delta_0}{\eta_0} + \frac{LV^2\eta_0}{2}\right) \cdot \frac{\ln T}{\sqrt{T} - 1}. \tag{6}$$

Theorem 3.3 establishes the convergence of Exp-Decay to a stationary point. With $\beta = \sqrt{T}$, Exp-Decay yields the rate of $\mathcal{O}(\ln T/\sqrt{T})$, which is comparable to the results in [28]. However, in [28], the $\mathcal{O}(\ln T/\sqrt{T})$-rate is obtained under the (arguably impractical) assumption that the initial step-size $\eta_0$ is bounded by $\mathcal{O}(1/\sqrt{T})$. This means that the initial step-size $\eta_0$ will be small if we plan on running the algorithm for a large number of iterations $T$. This is in conflict to the original motivation for using exponentially decaying step-sizes, namely to allow for large initial step-sizes which decrease as the algorithm progresses. On the other hand, our results in Theorem 3.3 only require that the initial step-size is bounded by $1/L$, which matches the largest fixed step-sizes that ensure convergence.

Another advantage of Theorem 3.3 is that it uses the output probability $P_t \propto 1/\eta_t$, instead of $P_t \propto \eta_t$ as in [28]. As discussed in the beginning of the section (and in Figure 1), $P_t \propto 1/\eta_t$ means that the output is much more likely to be chosen during the final iterates, whereas with $P_t \propto \eta_t$ the output is more likely to come from the initial iterates. Therefore, Theorem 3.3 better reflects the actual convergence of the algorithm, since in practice it is typically the final iterate that is used as the trained model. We illustrate this better in our experiments in Section 5.

### 3.3 Improved Convergence for the $1/\sqrt{t}$ Step-Size

The next theorem demonstrates how the idea of using the output distribution $P_t \propto 1/\eta_t$ instead of $P_t \propto \eta_t$ can improve standard convergence bounds for $1/\sqrt{t}$ step-sizes.

**Theorem 3.4.** *Suppose that the objective function is $L$-smooth and there exists a constant $\Delta_0 > 0$ such that $\mathbb{E}[f(x_t) - f^*] \leq \Delta_0$ for all $t \geq 1$, and that the stochastic gradient oracle is variance-bounded by $V^2$. If $\eta_t = \eta_0/\sqrt{t} \leq 1/L$, then*

$$\mathbb{E}[\|\nabla f(\hat{x}_T)\|^2] \leq \left(\frac{3\Delta_0}{\eta_0} + \frac{3LV^2\eta_0}{2}\right) \cdot \frac{1}{\sqrt{T}}$$

*where $\hat{x}_T$ is randomly drawn from $\{x_t\}_{t=1}^T$ with probabilities $P_t = \frac{1/\eta_t}{\sum_{t=1}^T 1/\eta_t}$.*

Theorem 3.4 establishes an $\mathcal{O}(1/\sqrt{T})$ convergence rate of SGD with $1/\sqrt{t}$ step-sizes. This improves the best known convergence bounds $\mathcal{O}(\ln T/\sqrt{T})$ for the algorithm, see Proposition 3.1 or [12].

## 4  Non-asymptotic Convergence for Strongly Convex Problems

We now investigate the convergence of the step decay step-sizes in the strongly convex case. We allow general convex and closed constraint set $X \subset \mathbb{R}^d$ and possibly non-smooth convex problems. To capture this setting, Algorithm 1 needs to be adjusted by including a projection step and subgradients (see Algorithm 2 in supplementary material). We also give the convergence for the general convex case (see Appendix A).

## 4.1 Strongly Convex and $L$-Smooth Functions

We first consider the case when $f$ is strongly convex and $L$-smooth (with respect to $x^*$).

**Theorem 4.1.** *Assume that the objective function $f$ is $\mu$-strongly convex on $\mathcal{X}$ and the stochastic gradient oracle is bounded by $G^2$. If we consider Algorithm 2 with $T > 1$, $N = \log_\alpha T$, $x_1^1 \in \mathcal{X}$, and $\eta_0 < 1/(2\mu)$, then we have*

$$\mathbb{E}\left\|x_{S+1}^N - x^*\right\|^2 \leq \frac{R}{\exp\left(A_3 \frac{T-1}{\ln T}\right)} + \frac{\alpha\eta_0 G^2 \exp\left(\frac{A_3}{\ln T}\right)}{2\mu A_3} \frac{\ln T}{T}$$

*where $A_3 = 2\mu\eta_0\alpha\ln\alpha/(\alpha - 1)$ and $R = \left\|x_1^1 - x^*\right\|^2$. Further, if the objective function $f$ is $L$-smooth with respect to $x^*$ then we can bound the objective function as follows*

$$\mathbb{E}[f(x_{S+1}^N) - f(x^*)] \leq \frac{L}{2}\mathbb{E}\left\|x_{S+1}^N - x^*\right\|^2.$$

The theorem provides an $\mathcal{O}(\ln T/T)$ theoretical guarantee for the last iterate under step decay step-size.[2] This bound matches the convergence rate of [11] for strongly convex least squares problems. Therefore, Theorem 4.1 can be considered a generalization of the result in [11] to general smooth and strongly convex problems.

From Theorem 4.1, we see that the effect of the initial condition $\|x_1^1 - x^\star\|^2$ is forgotten exponentially fast, unlike for $1/t$ step-sizes (e.g.[31]). For $1/t$ step-sizes, the initial learning rate is critical (both in theory and practice) to the convergence rate of SGD [32, 31]. For example, an initial step-size of $\eta_0 < 2/\mu$, will lead to a $t^{-\mu\eta_0/2}$ rate, far from the optimal rate $1/t$. For step decay, the initial step-size only affects the constant in front of the theoretical convergence bound and is not sensitive to the choice of initial step-size $\eta_0 < 1/(2\mu)$. We also observe this property of step-decay step-size in the experiments in Figure 4(b).

Under some step-sizes, e.g. $\eta_t = 1/t$, it is possible to get an $\mathcal{O}(1/T)$ convergence rate for SGD for smooth and strongly convex problems [31, 36, 19]. However, our next result shows that the rate in Theorem 4.1 is tight for Algorithm 2.

**Theorem 4.2** (Lower Bound). *Consider Algorithm 2 with $S = T/\log_\alpha T$ and $x_1^1 = 0$. For any $T \in \mathbb{N}^+$ and $\delta \in (0, 1)$, there exists a function $\tilde{f}_T : \mathcal{X} \to \mathbb{R}$, where $\mathcal{X} = [-4, 4]$, that is both 1-strongly convex and 1-smooth such that*

$$\tilde{f}_T(x_{S+1}^N) - \tilde{f}_T(x^*) \geq \frac{\ln(1/\delta)}{9\exp(2)\ln\alpha} \cdot \frac{\ln T}{T}$$

*with probability at least $\delta$, where $x^* = \min_{x\in\mathcal{X}} \tilde{f}_T(x)$.*

The high probability lower bound $\mathcal{O}(\ln T/T)$ matches the upper bound in Theorem 4.1. This suggests that the bound in Theorem 4.1 is actually tight for Algorithm 2 in the smooth strongly-convex case.

## 4.2 Strongly Convex and Non-Smooth Functions

For general, not necessarily smooth, strongly convex functions we have the following result.

**Theorem 4.3.** *Suppose that the objective function $f$ is $\mu$-strongly convex on $\mathcal{X}$ and the stochastic gradient oracle is bounded by $G^2$. If we run Algorithm 2 with $T > 1$, $S = T/\log_\alpha T$, $x_1^1 \in \mathcal{X}$, and $\eta_0 < 1/(2\mu)$, then we have*

$$\mathbb{E}[f(x_S^N) - f(x^*)] \leq \frac{R\ln T}{A_4 \exp\left(B_4\frac{T-\alpha}{\ln T}\right)} + C_4\frac{\ln T + 2}{T} + D_4\exp\left(\frac{E_4}{\ln T}\right)\frac{\ln^2 T}{T}$$

*where $A_4 = \eta_0\alpha\ln\alpha$, $B_4 = 2\mu A_4/(\alpha - 1)$, $C_4 = G^2\eta_0\alpha$, $D_4 = G^2/(2\mu\ln\alpha B_4)$, $E_4 = \alpha B_4$.*

The theorem shows that even without the smoothness assumption, we can still ensure convergence at the rate $\mathcal{O}(\ln^2 T/T)$. We can improve the rate to $\mathcal{O}(\ln T/T)$ with an averaging technique, as illustrated next. Moreover, the averaging technique improves the dependence of the parameter $\mu$ from $1/\mu^2$ (see Theorem 4.3) to $1/\mu$.

---

[2]Note that $\exp(A_3/\ln T) \leq \exp(A_3/\ln 2)$ for all $T \geq 2$ and $\exp(A_3/\ln T)$ converges to 1 as $T$ goes to $\infty$.

**Theorem 4.4.** *Under the assumptions of Theorem 4.3*

$$\mathbb{E}[f(\hat{x}_T) - f(x^*)] \leq \frac{A_5 R}{\exp\left(B_4 \frac{T}{\ln T} - 1\right)} + C_5 \frac{\ln T}{T}$$

*where* $\hat{x}_T = \sum_{t=t^*}^N \eta_t \sum_{i=1}^S x_i^t / (S \sum_{t=t^*}^N \eta_t)$, $t^* := \max\{0, \lfloor \log_\alpha(\eta_0 \alpha A_5 T / \log_\alpha T) \rfloor\}$, $A_5 = 2\mu\alpha/(\alpha - 1)$, $C_5 = \alpha(2 + 1/(\alpha^2 - 1))G^2/(2\mu \ln \alpha)$, *and* $B_4$ *is as defined in Theorem 4.3.*

## 5 Numerical Experiments

In this section, we evaluate the practical performance of step decay step-size and compare it against the following popular step-size policies: 1) constant step-size, $\eta_t = \eta_0$; 2) $1/t$ step-size, $\eta_t = \eta_0/(1 + a_0 t)$; 3) $1/\sqrt{t}$ step-size $\eta_t = \eta_0/(1 + a_0\sqrt{t})$; 4) Exp-Decay [28], $\eta_t = \eta_0/\alpha^t$ with $\alpha = (\beta/T)^{-1/T}$ for $\beta \geq 1$. In each experiment, we perform a grid search to select the best values for the free parameters $\eta_0, a_0, \beta$ as well as for the step decay step-size parameter $\alpha$. More details about the relationship between the different step-size policies are given in the supplementary material.

### 5.1 Experiments on MNIST with Neural Networks

Consider the classification task on MNIST database of handwritten digits using a fully-connected 2-layer neural network with 100 hidden nodes (784-100-10) [3]. We first explore the output $x_T$ at the last iterate $T$, which is the output of the algorithm that is typically used in practice. The theoretical output $\hat{x}_T$ (in Theorem 3.2) is drawn from all the previous iterates $\{x_i^t\}$ with probability $P_t \propto 1/\eta_t$. To get an insight into the relationship between the last iterate and theoretical output in Theorem 3.2, we randomly choose 6000 iterates (10% of the total iterates) with probability $P_t \propto 1/\eta_t$, record their exact training loss and testing loss, and calculate the probabilities (shown in Figure 1, middle and right). We can see that the theoretical output can reach the results of the last iterate with high probability, no matter training loss or testing loss.

In the same way, in order to show the advantages of probability $P_t \propto 1/\eta_t$ over $P_t \propto \eta_t$, we also implement Step-Decay with probability $P_t \propto \eta_t$ in Figure 1. We can see that the output with probability $P_t \propto 1/\eta_t$ is more concentrated at the last phase and has a higher probability, especially, in terms of loss, compared to the result of probability $P_t \propto \eta_t$.

The performance of the various step-size schedules is shown in Figure 2. Exp-Decay and Step-Decay perform better than other step-sizes both in loss (training and testing) and testing accuracy. Step-Decay has an advantage over Exp-Decay in the later stages of training where it attains a lower training and testing loss.

### 5.2 Experiments on CIFAR10 and CIFAR100

To illustrate the practical implications of the step decay step-size, we perform experiments with deep learning tasks on the CIFAR [1]. We will focus on results for CIFAR100 here, and present complementary results for CIFAR10 in the supplementary material. To eliminate the influence of stochasticity, all experiments are repeated 5 times.

We consider the benchmark experiments for CIFAR100 on a 100-layer DenseNet [22]. We employ vanilla SGD without dampening and use a weight decay of 0.0005. The optimal step-size and algorithm parameters are selected using a grid search detailed in the supplementary material. The results are shown in Figure 3. We observe that Step-Decay achieves the best results in terms of both testing loss and testing accuracy, and that it is also fast in reaching a competitive solution. Another observation is that as the iterates proceeds in each phase, its testing loss and accuracy is getting worse because its generalization ability is weakened. Therefore, deciding when to stop the iteration or reduce the step-size is important.

Finally, we compare the performance of Exp-Decay and Step-Decay on Nesterov's accelerated gradient (NAG) [34, 41] and other adaptive gradient methods, including AdaGrad [10], Adam [23] and AdamW [30]. The results are shown in Table 1. All the parameters involved in step-sizes, algorithms, and models are best-tuned (shown in supplementary material). The $\pm$ shows 95% confidence intervals of the mean accuracy value over 5 runs. We can see that compared to the

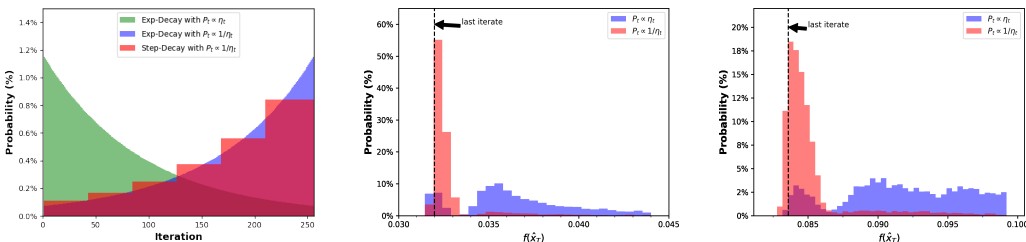

Figure 1: Output probability distributions (left); training loss (middle) and testing loss (right)

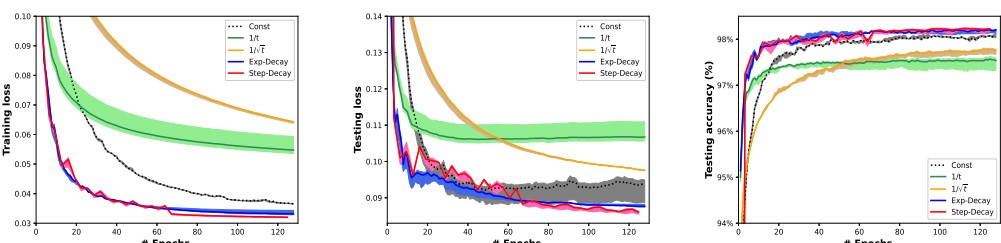

Figure 2: Results on MNIST: left - training loss; middle - testing loss; right - testing accuracy

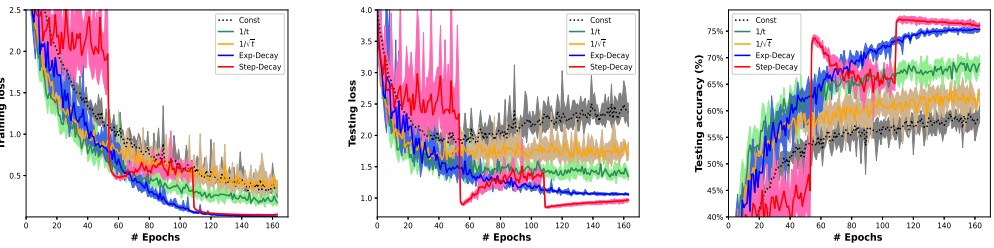

Figure 3: Results on CIFAR100: left - training loss; middle - testing loss; right - testing accuracy

Exp-Decay step-size, Step-Decay can reach higher testing accuracy on Adam, AdamW and NAG. This suggest that the Step-Decay step-size can be extended to other methods.

## 5.3 Experiments on Regularized Logistic Regression

We now turn our attention to how the step-decay step-size and other related step-sizes behave in the strongly convex setting. We consider the regularized logistic regression problem on the binary classification dataset rcv1.binary ($n = 20242$; $d = 47236$) from LIBSVM [2], where a 0.75 partition is used for training and the rest is for testing.

Figure 4(a) shows how a constant step-size has to be well-tuned to give good performance: if we choose it too small, convergence will be painstakingly slow; if we set it too large, iterates will stagnate or even diverge. This effect is also visualized in blue in Figure 4(b), where only a narrow range of values results in a low training loss for the constant step-size. In contrast, the initial step-size of both Exp-Decay and Step-Decay can be selected from a wide range ($\approx 10 - 10^4$) and still yield good results in the end. In other words, Exp-Decay and Step-Decay are more robust to the choice of initial step-size than the alternatives. A more thorough evaluation of all the considered step-sizes on logistic regression can be found in Supplementary D.3.

Table 1: Test accuracy on CIFAR100.

| Method | CIFAR100-DenseNet Testing accuracy |
|---|---|
| AdaGrad | $0.6197 \pm 0.00518$ |
| Adam + Exp-Decay | $0.6936 \pm \mathbf{0.00483}$ |
| Adam + Step-Decay | $\mathbf{0.7041} \pm 0.00971$ |
| AdamW + Exp-Decay | $0.7165 \pm 0.00353$ |
| AdamW + Step-Decay | $\mathbf{0.7335} \pm \mathbf{0.00261}$ |
| NAG + Exp-Decay | $0.7531 \pm 0.00606$ |
| NAG + Step-Decay | $\mathbf{0.7568} \pm \mathbf{0.00156}$ |

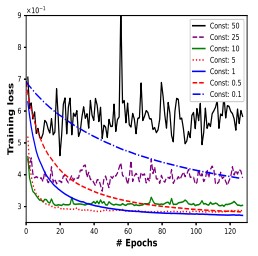
(a) Constant step-size

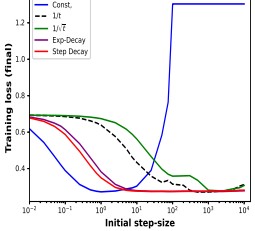
(b) Initial step-sizes

Figure 4: The results on rcv1.binary - logistic($L_2$)

# 6 Conclusion

We have provided theoretical guarantees for SGD under the step decay family of step-sizes, widely used in deep learning. Our first results established a near-optimal $\mathcal{O}(\ln T/\sqrt{T})$ rate for step decay step-sizes in the non-convex setting. A key step in our analysis was to use a novel non-uniform probability distribution $P_t \propto 1/\eta_t$ for selecting the output of the algorithm. We showed that this approach allows to improve the convergence results for SGD under other step-sizes as well, e.g., by removing the $\ln T$ term in the best known convergence rate for $\eta_t = 1/\sqrt{t}$ step-sizes. Moreover, we established near-optimal (compared to the min-max rate) convergence rates for general convex, strongly convex and smooth, and strongly convex and non-smooth problems. We illustrated the superior performance of step-decay step-sizes for training of large-scale deep neural networks. In the experiments, we observed that as the iterates proceeding in each phase, their generalization abilities are getting worse. Therefore, it will be an interesting to study how to best select the size of inner-loop $S$ (instead of constant or exponentially growing) to avoid the loss of generalization.

## Broader Impact

This is mostly a theoretical work and, therefore, it does not have any direct negative societal impacts.

## Acknowledgements and Disclosure of Funding

This work was supported partially by the Wallenberg Artificial Intelligence, Autonomous Systems and Software Program (WASP) funded by Knut and Alice Wallenberg Foundation, and the Swedish Research Council under contract 2019-05319 and 2020-03607.

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
