7: $\qquad x_{i+1}^t = \Pi_{\mathcal{X}}(x_i^t - \eta_t \hat{g}_i^t)$, where $\Pi_{\mathcal{X}}$ is the projection operator on $\mathcal{X}$
8: $\quad$ **end for**
9: $\quad x_1^{t+1} = x_{S+1}^t$
10: **end for**
11: **Return:** $x_{S+1}^N$

---

## A  Non-asymptotic Convergence for General Convex Problems

We now establish the first convergence rate results for the step decay step-size in the general convex setting. More specifically, we consider a possibly non-differentiable convex objective function on a closed and convex constraint set $\mathcal{X} \subseteq \mathbb{R}^d$. For this problem class, we analyze the projected SGD with step decay step-size detailed in Algorithm 2.

We have the following convergence guarantees:

**Theorem A.1.** *Suppose that the objective function $f$ is convex on $\mathcal{X}$ and $\sup_{x,y \in \mathcal{X}} \|x - y\|^2 \leq D^2$. The stochastic gradient oracle is bounded by $G^2$. If we run Algorithm 2 with $T > 1$, $S = 2T/\log_\alpha T$, $x_1^1 \in \mathcal{X}$, then we have*

$$\frac{1}{S} \sum_{i=1}^{S} \mathbb{E}[f(x_i^N)] - f^* \leq A_2 \frac{\ln T}{\sqrt{T}} + \frac{B_2}{\sqrt{T}},$$

*where $A_2 = D^2/(4\eta_0 \alpha \ln \alpha)$ and $B_2 = G^2 \eta_0 \alpha/2$. Moreover, we have the following bound on the final iterate:*

$$\mathbb{E}[f(x_S^N)] - f^* \leq (A_2 + B_2) \frac{\ln T}{\sqrt{T}} + \frac{(2 + \ln 2)B_2}{\sqrt{T}}.$$

Theorem A.1 establishes an $\mathcal{O}(\ln T/\sqrt{T})$ convergence rate for both the average objective function value and the objective function value at the final iterate. This convergence rate is comparable to the results obtained for other diminishing step-sizes such as $\eta_t = \eta_0/\sqrt{t}$ under the same assumptions [38].

*Proof.* (of **Theorem A.1**) The convexity of $f$ yields $\langle g_i^t, x - x_i^t \rangle \leq f(x) - f(x_i^t)$ for any $x \in \mathcal{X}$, where $g_i^t \in \partial f(x_i^t)$. Also, by convexity of $\mathcal{X}$, we have $\|\Pi_{\mathcal{X}}(u) - v\| \leq \|u - v\|$ for any points $u \in \mathbb{R}^d$ and $v \in \mathcal{X}$. Using these inequalities and applying the assumption that the stochastic gradient oracle is bounded by $G^2$, i.e., $\mathbb{E}[\|\hat{g}_i^t\|^2] \leq G^2$, for any $x \in \mathcal{X}$, we have

$$\mathbb{E}[\|x_{i+1}^t - x\|^2] = \mathbb{E}[\|\Pi_{\mathcal{X}}(x_i^t - \eta_t \hat{g}_i^t) - x\|^2] \leq \mathbb{E}[\|x_i^t - \eta_t \hat{g}_i^t - x\|^2]$$
$$\leq \mathbb{E}[\|x_i^t - x\|^2] - 2\eta_t \mathbb{E}[\langle g_i^t, x_i^t - x \rangle] + \eta_t^2 G^2$$
$$\leq \mathbb{E}[\|x_i^t - x\|^2] - 2\eta_t [f(x_i^t) - f(x)] + \eta_t^2 G^2. \tag{7}$$

Shifting $[f(x_i^t) - f(x)]$ to the left side gives

$$2\eta_t \mathbb{E}[f(x_i^t) - f(x)] \leq \mathbb{E}[\|x_i^t - x\|^2] - \mathbb{E}[\|x_{i+1}^t - x\|^2] + \eta_t^2 G^2. \tag{8}$$

Now, consider the final phase $N$ (let $t = N$) and $x = x^*$ and apply (8) recursively from $i = 1$ to $S$ to obtain

$$2\eta_N \sum_{i=1}^{S} \mathbb{E}[f(x_i^N) - f^*] \leq \mathbb{E}[\|x_1^N - x^*\|^2] + S\eta_N^2 G^2. \tag{9}$$

Combining the assumption that $\sup_{x,y \in \mathcal{X}} \|x - y\|^2 \leq D^2$ for some finite $D$ with the expression for the step-size in the final phase that $\eta_N = \eta_0 \alpha / \sqrt{T}$, we have

$$\frac{\sum_{i=1}^{S} \mathbb{E}[f(x_i^N)]}{S} - f^* \leq \frac{\mathbb{E}\left[\|x_1^N - x^*\|^2\right]}{2\eta_N S} + \frac{G^2 \eta_N}{2}$$

$$\leq \frac{D^2}{4\eta_0 \alpha} \cdot \frac{\log_\alpha T}{\sqrt{T}} + \frac{G^2 \eta_0 \alpha}{2} \cdot \frac{1}{\sqrt{T}}. \tag{10}$$

We have thus proven the first part of Theorem A.1. Next, based on the above results, we prove the error bound for the last iterate. We focus on the last phase $N$ and apply (8) recursively from $i = S - k$ to $S$ to find

$$2\eta_N \sum_{i=S-k}^{S} \mathbb{E}[f(x_i^N) - f(x_{S-k}^N)] \leq \mathbb{E}\left[\|x_{S-k}^N - x_{S-k}^N\|^2\right] - \mathbb{E}\left[\|x_{S+1}^N - x_{S-k}^N\|^2\right] + (k+1)\eta_N^2 G^2. \tag{11}$$

Introducing $W_{k+1}^N := \frac{1}{k+1} \sum_{i=S-k}^{S} \mathbb{E}[f(x_i^N)]$, inequality (11) implies that

$$-f(x_{S-k}^N) \leq -W_{k+1}^N + \frac{\eta_N G^2}{2}. \tag{12}$$

By the definition of $W_{k+1}^N$, we have $(k+1)W_{k+1}^N - kW_k^N = f(x_{S-k}^N)$. Using this formula and applying (12) gives

$$kW_k^N = (k+1)W_{k+1}^N - f(x_{S-k}^N) \leq (k+1)W_{k+1}^N - W_{k+1}^N + \frac{\eta_N G^2}{2}.$$

Dividing by $k$, we get

$$W_k^N \leq W_{k+1}^N + \frac{\eta_N G^2}{2k}.$$

Using the above inequality repeatedly for $k = 1, 2, \cdots, S - 1$, we have

$$W_1^N \leq W_S^N + \frac{\eta_N G^2}{2} \sum_{k=1}^{S-1} \frac{1}{k} \leq W_S^N + \frac{G^2 \eta_0 \alpha}{2\sqrt{T}} \left(\ln\left(\frac{2T}{\log_\alpha T}\right) + 1\right). \tag{13}$$

Recalling the definition of $W_S^N$ and applying (10) into the above inequality, we obtain

$$\mathbb{E}[f(x_S^N)] - f^* \leq \frac{D^2}{4\eta_0 \alpha} \cdot \frac{\log_\alpha T}{\sqrt{T}} + \frac{G^2 \eta_0 \alpha}{2} \cdot \frac{\ln T}{\sqrt{T}} + \frac{G^2 \eta_0 \alpha (1 + \ln(2)/2)}{\sqrt{T}}.$$

The proof is complete. $\qquad\square$

# B   Proofs of Section 3

Before presenting the proofs of Section 3, we state and prove the following useful lemma.

**Lemma B.1.** *Suppose that $f$ is $L$-smooth on $\mathbb{R}^d$ and the stochastic gradient oracle is variance-bounded by $V^2$. If $\eta_t \leq 1/L$, consider the SGD algorithm, we have*

$$\frac{\eta_t}{2} \mathbb{E}[\|\nabla f(x_t)\|^2] \leq \mathbb{E}[f(x_t)] - \mathbb{E}[f(x_{t+1})] + \frac{LV^2 \eta_t^2}{2}.$$

*Proof.* Recall the SGD iterations $x_{t+1} = x_t - \eta_t \hat{g}_t$, where $\mathbb{E}[\hat{g}_t] = \nabla f(x_t)$. By the smoothness of $f$ on $\mathbb{R}^d$, we have

$$f(x_{t+1}) \leq f(x_t) + \langle \nabla f(x_t), x_{t+1} - x_t \rangle + \frac{L}{2} \|x_{t+1} - x_t\|^2$$

$$\leq f(x_t) + \langle \nabla f(x_t), -\eta_t \hat{g}_t \rangle + \frac{L}{2} \|x_{t+1} - x_t\|^2$$

$$\leq f(x_t) - \eta_t \langle \nabla f(x_t), \hat{g}_t \rangle + \frac{L\eta_t^2}{2} \|\hat{g}_t\|^2. \tag{14}$$

The stochastic gradient oracle is variance-bounded by $V^2$, i.e., $\mathbb{E}[\|\hat{g}_t - \nabla f(x_t)\|^2] \leq V^2$. Taking expectation on both sides of (14) and applying $\mathbb{E}[\hat{g}_t] = \nabla f(x_t)$ and the variance-bounded assumption gives

$$\mathbb{E}[f(x_{t+1})] \leq \mathbb{E}[f(x_t)] - \eta_t \mathbb{E}[\|\nabla f(x_t)\|^2] + \frac{L\eta_t^2}{2}\mathbb{E}[\|\hat{g}_t\|^2]$$

$$\leq \mathbb{E}[f(x_t)] - \eta_t \mathbb{E}[\|\nabla f(x_t)\|^2] + \frac{L\eta_t^2}{2}\mathbb{E}[\|\hat{g}_t - \nabla f(x_t) + \nabla f(x_t)\|^2]$$

$$\leq \mathbb{E}[f(x_t)] - \eta_t \mathbb{E}[\|\nabla f(x_t)\|^2] + \frac{L\eta_t^2}{2}\left(\mathbb{E}[\|\hat{g}_t - \nabla f(x_t)\|^2] + \mathbb{E}[\|\nabla f(x_t)\|^2]\right)$$

$$\leq \mathbb{E}[f(x_t)] + \left(-\eta_t + \frac{L\eta_t^2}{2}\right)\mathbb{E}[\|\nabla f(x_t)\|^2] + \frac{L\eta_t^2}{2}\mathbb{E}[\|\hat{g}_t - \nabla f(x_t)\|^2]$$

$$\leq \mathbb{E}[f(x_t)] + \left(-\eta_t + \frac{L\eta_t^2}{2}\right)\mathbb{E}[\|\nabla f(x_t)\|^2] + \frac{LV^2\eta_t^2}{2}. \tag{15}$$

where the second inequality follows the fact that gradient $\hat{g}_t$ is unbiased ($\mathbb{E}[\hat{g}_t] = \nabla f(x_t)$). We thus have that

$$\mathbb{E}[\|\hat{g}_t - \nabla f(x_t) + \nabla f(x_t)\|^2] = \mathbb{E}[\|\hat{g}_t - \nabla f(x_t)\|^2 + \|\nabla f(x_t)\|^2 + 2\langle \hat{g}_t - \nabla f(x_t), \nabla f(x_t)\rangle]$$

$$= \mathbb{E}[\|\hat{g}_t - \nabla f(x_t)\|^2] + \mathbb{E}[\|\nabla f(x_t)\|^2].$$

Since $\eta_0 \leq 1/L$, we have $-\eta_t + \eta_t^2 L/2 \leq -\eta_t/2$. Using this inequality, shifting $\mathbb{E}[\|\nabla f(x_t)\|^2]$ to the left side, and re-arranging (15) gives

$$\frac{\eta_t}{2}\mathbb{E}[\|\nabla f(x_t)\|^2] \leq \mathbb{E}[f(x_t)] - \mathbb{E}[f(x_{t+1})] + \frac{LV^2\eta_t^2}{2}.$$

$$\square$$

*Proof.* (of **Proposition 3.1**) By Lemma B.1, the following inequality holds for SGD

$$\frac{\eta_t}{2}\mathbb{E}[\|\nabla f(x_t)\|^2] \leq \mathbb{E}[f(x_t)] - \mathbb{E}[f(x_{t+1})] + \frac{LV^2\eta_t^2}{2}. \tag{16}$$

Under the diminishing step-size $\eta_t = \eta_0/\sqrt{t}$, we estimate the summation of $\eta_t$ and $\eta_t^2$ from $t = 1$ to $T$, respectively:

$$\sum_{t=1}^{T} \eta_t = \eta_0 \sum_{t=1}^{T} \frac{1}{\sqrt{t}} \geq \eta_0 \int_{t=1}^{T} \frac{1}{\sqrt{t}} dt = 2\eta_0(\sqrt{T} - 1)$$

$$\sum_{t=1}^{T} \eta_t^2 = \eta_0^2 \sum_{t=1}^{T} \frac{1}{t} \leq \eta_0^2\left(1 + \int_{t=1}^{T} \frac{1}{t} dt\right) = \eta_0^2(\ln T + 1).$$

Recall that the output $\hat{x}_T$ is chosen randomly from the sequence $\{x_t\}_{t=1}^{T}$ with probability $P_t = \frac{\eta_t}{\sum_{t=1}^{T} \eta_t}$. We thus have

$$\mathbb{E}[\|\nabla f(\hat{x}_T)\|] = \frac{\eta_t \mathbb{E}[\|\nabla f(x_t)\|^2]}{\sum_{t=1}^{T} \eta_t}$$

$$\leq \frac{2\sum_{t=1}^{T} \mathbb{E}[f(x_t) - \mathbb{E}[f(x_{t+1})]]}{\sum_{t=1}^{T} \eta_t} + \frac{LV^2 \sum_{t=1}^{T} \eta_t^2}{\sum_{t=1}^{T} \eta_t}$$

$$\leq \frac{2(f(x_1) - f^*)}{\sum_{t=1}^{T} \eta_t} + \frac{LV^2 \sum_{t=1}^{T} \eta_t^2}{\sum_{t=1}^{T} \eta_t}$$

$$\leq \frac{f(x_1) - f^*}{\eta_0(\sqrt{T} - 1)} + \frac{LV^2 \eta_0(\ln T + 1)}{2(\sqrt{T} - 1)}.$$

$$\square$$

*Proof.* (of **Theorem 3.2**) Invoking the result of Lemma B.1, at the current iterate $x_i^t$, the following inequality holds:

$$\frac{\eta_t}{2}\mathbb{E}[\|\nabla f(x_i^t)\|^2] \leq \mathbb{E}[f(x_i^t)] - \mathbb{E}[f(x_{i+1}^t)] + \frac{LV^2\eta_t^2}{2},$$

Dividing both sides by $\eta_t^2/2$ yields

$$\frac{1}{\eta_t}\mathbb{E}[\|\nabla f(x_i^t)\|^2] \leq \frac{2}{\eta_t^2}\left(\mathbb{E}[f(x_i^t)] - \mathbb{E}[f(x_{i+1}^t)]\right) + LV^2 \tag{17}$$

At each inner phase for $i \in [S]$, the step-size $\eta_t$ is a constant. Applying (17) repeatedly for $i = 1, 2, \cdots S$ gives

$$\frac{1}{\eta_t}\sum_{i=1}^{S}\mathbb{E}[\|\nabla f(x_i^t)\|^2] \leq \frac{2}{\eta_t^2}\left(\mathbb{E}[f(x_1^t)] - \mathbb{E}[f(x_{S+1}^t)]\right) + LV^2 \cdot S \tag{18}$$

Since the output $\hat{x}_T$ of Algorithm 1 is randomly chosen from all previous iterates $\{x_i^t\}$ with probability $P_i^t = \frac{1/\eta_t}{S\sum_{l=1}^{N}1/\eta_t}$, we have

$$\mathbb{E}[\|\nabla f(\hat{x}_T)\|^2] = \frac{1}{S\sum_{t=1}^{N}(1/\eta_t)}\sum_{t=1}^{N}\frac{1}{\eta_t}\sum_{i=1}^{S}\|\nabla f(x_i^t)\|^2$$

$$\leq \frac{1}{S\sum_{t=1}^{N}(1/\eta_t)}\sum_{t=1}^{N}\frac{2}{\eta_t^2}\left(\mathbb{E}[f(x_1^t)] - \mathbb{E}[f(x_{S+1}^t)]\right) + \frac{LV^2 \cdot S}{S\sum_{t=1}^{N}1/\eta_t}$$

$$\leq \frac{2\eta_0}{S\eta_0^2\sum_{t=1}^{N}\alpha^{t-1}}\sum_{t=1}^{N}\alpha^{2(t-1)}\left(\mathbb{E}[f(x_1^t)] - \mathbb{E}[f(x_{S+1}^t)]\right) + \frac{LV^2\eta_0}{\sum_{t=1}^{N}\alpha^{t-1}}. \tag{19}$$

By the update rule in Algorithm 1, the last point $x_{S+1}^t$ of the $t^{\text{th}}$ loop is the starting point of the next loop, i.e., $x_{S+1}^t = x_1^{t+1}$ for each $t \in [N]$. Applying the bounded assumption on the objective function that $\mathbb{E}[f(x_1^t) - f^*] \leq \Delta_0$ for all $t \geq 1$,

$$\sum_{t=1}^{N}\alpha^{2(t-1)}\left(\mathbb{E}[f(x_1^t)] - \mathbb{E}[f(x_{S+1}^t)]\right) \leq f(x_1^1) - f^* + (\alpha^2 - 1)\sum_{t=2}^{N}\alpha^{2(t-2)} \cdot \mathbb{E}[f(x_1^t) - f^*]$$

$$\leq \alpha^{2(N-1)}\Delta_0.$$

Plugging this inequality into (19) and $S = T/N$ we obtain that

$$\mathbb{E}[\|\nabla f(\hat{x}_T)\|^2] \leq \frac{2\eta_0\alpha^{2(N-1)}\Delta_0}{\eta_0^2 S\sum_{t=1}^{N}\alpha^{t-1}} + \frac{LV^2\eta_0}{\sum_{t=1}^{N}\alpha^{t-1}}$$

$$\leq \frac{2(\alpha - 1)\Delta_0}{\alpha^2\eta_0} \cdot \frac{N\alpha^N}{T(1 - \alpha^{-N})} + \frac{\eta_0 LV^2(\alpha - 1)}{\alpha^N - 1}.$$

Substituting $N = (\log_\alpha T)/2$ and $S = 2T/\log_\alpha T$ into the above inequality gives

$$\mathbb{E}[\|\nabla f(\hat{x}_T)\|^2] \leq \frac{(\alpha - 1)\Delta_0}{\eta_0\alpha^2} \cdot \frac{\log_\alpha T}{\sqrt{T} - 1} + \frac{\eta_0 LV^2(\alpha - 1)}{\sqrt{T} - 1}.$$

Then by changing the base $\alpha$ of $\log_\alpha T$ to be natural logarithm, the theorem is proved. $\qquad\square$

*Proof.* (of **Theorem 3.3**) Based on Lemma B.1, at the current iterate $x_t$, we have

$$\frac{\eta_t}{2}\mathbb{E}[\|\nabla f(x_t)\|^2] \leq \mathbb{E}[f(x_t)] - \mathbb{E}[f(x_{t+1})] + \frac{LV^2\eta_t^2}{2}.$$

Dividing the above inequality by $\eta_t^2/2$ and summing over $t = 1$ to $T$ gives

$$\sum_{t=1}^{T}\frac{1}{\eta_t}\mathbb{E}[\|\nabla f(x_t)\|^2] \leq \sum_{t=1}^{T}\frac{2(\mathbb{E}[f(x_t)] - \mathbb{E}[f(x_{t+1})])}{\eta_t^2} + LV^2 T \tag{20}$$

Applying the assumption that the objective function $\mathbb{E}[f(x_t) - f^*] \leq \Delta_0$ for all $t \geq 1$, and recalling the definition of the exponential decay step-size [28], i.e., $\eta_t = \eta_0/\alpha^t$ where $\alpha = (\beta/T)^{-1/T}$ and $\beta \geq 1$, we find

$$\sum_{t=1}^{T} \frac{2(\mathbb{E}[f(x_t)] - \mathbb{E}[f(x_{t+1})])}{\eta_t^2} \leq \frac{2(f(x_1) - f^*)}{\eta_1^2} + 2\sum_{t=2}^{T} \left(\frac{1}{\eta_t^2} - \frac{1}{\eta_{t-1}^2}\right) \mathbb{E}[f(x_t) - f^*]$$

$$\leq \frac{2(f(x_1) - f^*)}{\eta_1^2} + \frac{2}{\eta_0^2}\left(\alpha^2 - 1\right) \sum_{t=2}^{T} \alpha^{2(t-1)} \Delta_0$$

$$\leq \frac{2\Delta_0}{\eta_0^2} \cdot \left(\frac{T}{\beta}\right)^2. \tag{21}$$

Next, we estimate the sum of $1/\eta_t$ from $t = 1$ to $T$:

$$\sum_{t=1}^{T} \frac{1}{\eta_t} = \frac{1}{\eta_0}\sum_{t=1}^{T} \alpha^t = \frac{\alpha(1 - \alpha^T)}{\eta_0(1 - \alpha)} = \frac{(\frac{T}{\beta} - 1)}{\eta_0(1 - 1/\alpha)}$$

$$\geq \frac{(\frac{T}{\beta} - 1)}{\eta_0 \ln(\alpha)} = \frac{(\frac{T}{\beta} - 1)T}{\eta_0 \ln(\frac{T}{\beta})}. \tag{22}$$

where the last inequality follows from the fact that $1 - x \leq \ln(\frac{1}{x})$ for all $x > 0$.

Combining the selection rule for $\hat{x}_T$ with inequalities (20), (21) and (22), we have

$$\mathbb{E}[\|\nabla f(\hat{x}_T)\|^2] = \frac{1}{\sum_{t=1}^{T} 1/\eta_t}\left[\sum_{t=1}^{T} \frac{1}{\eta_t}\mathbb{E}[\|\nabla f(x_t)\|^2]\right]$$

$$\leq \frac{\eta_0 \ln(\frac{T}{\beta})}{(\frac{T}{\beta} - 1)T}\left[\frac{2\Delta_0}{\eta_0^2}\left(\frac{T}{\beta}\right)^2 + LV^2T\right].$$

Letting $\beta = \sqrt{T}$, we see that

$$\mathbb{E}[\|\nabla f(\hat{x}_T)\|^2] \leq \frac{\eta_0 \ln T}{2(\sqrt{T} - 1)}\left[\frac{2\Delta_0}{\eta_0^2} + LV^2\right],$$

which concludes the proof. $\qquad\square$

*Proof.* (of **Theorem 3.4**) In this case, we consider the diminishing step-size $\eta_t = \eta_0/\sqrt{t}$. By Lemma B.1:

$$\frac{\eta_t}{2}\mathbb{E}[\|\nabla f(x_t)\|^2] \leq \mathbb{E}[f(x_t)] - \mathbb{E}[f(x_{t+1})] + \frac{LV^2\eta_t^2}{2}. \tag{23}$$

In the same way as Theorems 3.2 and 3.3, we divide (23) by $\eta_t^2/2$ and sum over $t = 1$ to $T$ to obtain

$$\sum_{t=1}^{T} \frac{1}{\eta_t}\mathbb{E}[\|\nabla f(x_t)\|^2] \leq \sum_{t=1}^{T} \frac{2(\mathbb{E}[f(x_t)] - \mathbb{E}[f(x_{t+1})])}{\eta_t^2} + LV^2T$$

$$\leq \frac{2}{\eta_0^2}\left[f(x_1) - f^* + \sum_{t=2}^{T}(t - (t-1))\mathbb{E}[f(x_t) - f^*]\right] + LV^2T$$

$$\leq \frac{2T\Delta_0}{\eta_0^2} + LV^2T. \tag{24}$$

Recalling that the output $\hat{x}_T$ is randomly chosen from the sequence $\{x_t\}_{t=1}^{T}$ with probability $P_t = \frac{1/\eta_t}{\sum_{t=1}^{T} 1/\eta_t}$ and applying (24), yields

$$\mathbb{E}[\|\nabla f(\hat{x}_T)\|^2] = \frac{\sum_{t=1}^{T} 1/\eta_t \mathbb{E}[\|\nabla f(x_t)\|^2]}{\sum_{t=1}^{T} 1/\eta_t} \leq \frac{1}{\sum_{t=1}^{T} 1/\eta_t}\left[\frac{2T\Delta_0}{\eta_0^2} + LV^2T\right]$$

$$\leq \left(\frac{3\Delta_0}{\eta_0} + \frac{3LV^2\eta_0}{2}\right) \cdot \frac{1}{\sqrt{T}}.$$

where the last inequality holds because $\sum_{t=1}^{T} 1/\eta_t \geq 1/\eta_0 \cdot \int_{t=0}^{T} \sqrt{t} dt = \frac{2}{3\eta_0} T^{3/2}$. The proof is complete. $\qquad\square$

## C   Proofs of Section 4

*Proof.* (of **Theorem 4.1** ) In this case, we consider the step decay step-size (see Algorithm 2) with $S = T/\log_\alpha T$ and $N = \log_\alpha T$. By the $\mu$-strongly convexity of $f$ on $\mathcal{X}$, we have

$$f(x^*) \geq f(x_i^t) + \langle g_i^t, x^* - x_i^t \rangle + \frac{\mu}{2} \left\| x_i^t - x^* \right\|^2, \text{ and} \tag{25}$$

$$f(x_i^t) \geq f(x^*) + \langle g^*, x_i^t - x^* \rangle + \frac{\mu}{2} \left\| x_i^t - x^* \right\|^2, \forall g^* \in \partial f(x^*). \tag{26}$$

Due to the fact that $x^*$ minimizes $f$ on $\mathcal{X}$, we have $\langle g^*, x - x^* \rangle \geq 0$ for all $g^* \in \partial f(x^*)$ and $x \in \mathcal{X}$. In particular, for $x = x_i^t$ we have $\langle g^*, x_i^t - x^* \rangle \geq 0$. Plugging this into (26) and re-arranging (25) and (26) gives

$$\langle g_i^t, x_i^t - x^* \rangle \geq \frac{\mu}{2} \left\| x_i^t - x^* \right\|^2 + f(x_i^t) - f(x^*) \geq \mu \left\| x_i^t - x^* \right\|^2. \tag{27}$$

By the convexity of $\mathcal{X}$, we have $\left\| \Pi_{\mathcal{X}}(u) - v \right\|^2 \leq \left\| u - v \right\|^2$ for any $u \in \mathbb{R}^d$ and $v \in \mathcal{X}$. Then applying the update rule of Algorithm 2 and using these inequalities gives

$$
\begin{aligned}
\mathbb{E}\left[ \left\| x_{i+1}^t - x^* \right\|^2 \mid \mathcal{F}_i^t \right] &= \mathbb{E}\left[ \left\| \Pi_{\mathcal{X}}(x_i^t - \eta_t \hat{g}_i^t) - x^* \right\|^2 \mid \mathcal{F}_i^t \right] \leq \mathbb{E}\left[ \left\| x_i^t - \eta_t \hat{g}_i^t - x^* \right\|^2 \mid \mathcal{F}_i^t \right] \\
&\leq \left\| x_i^t - x^* \right\|^2 - 2\eta_t \mathbb{E}\left[ \langle \hat{g}_i^t, x_i^t - x^* \rangle \mid \mathcal{F}_i^t \right] + \eta_t^2 \mathbb{E}\left[ \left\| \hat{g}_i^t \right\|^2 \mid \mathcal{F}_i^t \right] \\
&\leq \left\| x_i^t - x^* \right\|^2 - 2\eta_t \langle g_i^t, x_i^t - x^* \rangle + \eta_t^2 G^2 \\
&\leq (1 - 2\mu\eta_t) \left\| x_i^t - x^* \right\|^2 + \eta_t^2 G^2, \tag{28}
\end{aligned}
$$

where the third inequality follows from the stochastic gradient oracle is bounded by $G^2$, i.e., $\hat{g}_i^t$ satisfies that $\mathbb{E}[\hat{g}_i^t] = g_i^t \in \partial f(x_i^t)$ and $\mathbb{E}[\|\hat{g}_i^t\|^2] \leq G^2$. In this case, the time horizon $T$ is divided into $N = \log_\alpha T$ phases and each is of length $S = T/\log_\alpha T$. Recursively applying (28) from $i = 1$ to $S$ in the $t^{\text{th}}$ phase and using the assumption $\eta_0 < 1/(2\mu)$ gives

$$
\begin{aligned}
\mathbb{E}\left[ \left\| x_{S+1}^t - x^* \right\|^2 \right] &\leq (1 - 2\mu\eta_t)^S \mathbb{E}\left[ \left\| x_1^t - x^* \right\|^2 \right] + G^2 \eta_t^2 \sum_{l=0}^{S-1} (1 - 2\mu\eta_t)^l \\
&\leq (1 - 2\mu\eta_t)^S \mathbb{E}\left[ \left\| x_1^t - x^* \right\|^2 \right] + \frac{G^2}{2\mu} \cdot \eta_t. \tag{29}
\end{aligned}
$$

Repeating the recursion (29) from $t = 1$ to $N$, we get

$$
\begin{aligned}
\mathbb{E}\left[ \left\| x_{S+1}^N - x^* \right\|^2 \right] &\leq \prod_{t=1}^{N} (1 - 2\mu\eta_t)^S \left\| x_1^1 - x^* \right\|^2 + \frac{G^2}{2\mu} \sum_{t=1}^{N} \eta_t \cdot \prod_{l>t}^{N} (1 - 2\mu\eta_l)^S \\
&\leq \exp\left( -2\mu S \sum_{t=1}^{N} \eta_t \right) \left\| x_1^1 - x^* \right\|^2 + \frac{G^2}{2\mu} \sum_{t=1}^{N} \eta_t \exp\left( -2\mu S \sum_{l>t}^{N} \eta_l \right), \tag{30}
\end{aligned}
$$

where the second inequality follows from the fact that $(1 + x)^s \leq \exp(sx)$ for any $x \in \mathbb{R}$ and $s > 0$. Recalling the formula for the step decay step-size, $\eta_t = \eta_0/\alpha^{t-1}$ in the $t^{\text{th}}$ phase, and that $N = \log_\alpha T$ and $S = T/\log_\alpha T$, we can estimate the two sums that appear in (30) as follows:

$$S \sum_{t=1}^{N} \eta_t = \frac{T}{\log_\alpha T} \cdot \frac{\eta_0(1 - \alpha^{-N})}{1 - 1/\alpha} = \frac{\eta_0 \alpha}{(\alpha - 1)} \cdot \frac{T - 1}{\log_\alpha T}$$

$$S \sum_{l>t}^{N} \eta_l = \frac{T}{\log_\alpha T} \cdot \frac{\eta_0 \alpha^{-t}(1 - \alpha^{-(N-t)})}{(1 - 1/\alpha)} = \frac{\eta_0 \alpha}{(\alpha - 1)} \cdot \frac{T\alpha^{-t} - 1}{\log_\alpha T}.$$

Using these inequalities in (30) gives

$$\mathbb{E}\left[\left\|x_{S+1}^N - x^*\right\|^2\right] \le \exp\left(-\frac{2\mu\eta_0\alpha}{\alpha-1}\cdot\frac{T-1}{\log_\alpha T}\right)\left\|x_1^1 - x^*\right\|^2 + \frac{G^2\eta_0}{2\mu}\sum_{t=1}^{N}\frac{1}{\alpha^{t-1}}\exp\left(-\frac{2\mu\eta_0\alpha}{\alpha-1}\cdot\frac{T\alpha^{-t}-1}{\log_\alpha T}\right).$$
(31)

Next, we turn to bound the right-hand side of (31). Let $t^* := \max\left\{0, \left\lfloor\log_\alpha\left(\frac{2\mu\eta_0\alpha}{\alpha-1}\cdot\frac{T}{\log_\alpha T}\right)\right\rfloor\right\}$. If $t^* \ge 1$, we prefer to divide the second term into two parts. First, we estimate this term for $t \le t^*$:

$$
\begin{aligned}
\frac{G^2\eta_0}{2\mu}\sum_{t=1}^{t^*}\frac{1}{\alpha^{t-1}}\exp\left(-\frac{2\mu\eta_0\alpha}{\alpha-1}\cdot\frac{T\alpha^{-t}-1}{\log_\alpha T}\right) &\le \frac{G^2\eta_0}{2\mu}\sum_{t=1}^{t^*}\frac{1}{\alpha^{t-1}}\exp\left(-\frac{\alpha^{t^*}}{\alpha^t} + \frac{2\mu\eta_0\alpha}{(\alpha-1)\log_\alpha T}\right) \\
&\le \frac{G^2\eta_0\alpha\exp\left(\frac{2\mu\eta_0\alpha}{(\alpha-1)\log_\alpha T}\right)}{2\mu\alpha^{t^*}}\cdot\sum_{t=1}^{t^*}\frac{\alpha^{t^*}}{\alpha^t}\exp\left(-\frac{\alpha^{t^*}}{\alpha^t}\right) \\
&\le \frac{G^2\eta_0\alpha\exp\left(\frac{2\mu\eta_0\alpha}{(\alpha-1)\log_\alpha T}-1\right)}{2\mu\alpha^{t^*}} \\
&\le \frac{G^2(\alpha-1)\exp\left(\frac{\mu\eta_0\alpha}{(\alpha-1)\log_\alpha T}-1\right)}{2\mu^2}\cdot\frac{\log_\alpha T}{T},
\end{aligned}
$$
(32)

where the third inequality uses that $\int_{x=1}^{+\infty} x\exp(-x)dx \le 2/\exp(1)$. Next, we estimate the term for $t^* < t \le N$:

$$
\begin{aligned}
\frac{G^2\eta_0}{2\mu}\sum_{t=t^*+1}^{N}\frac{1}{\alpha^{t-1}}\exp\left(-\frac{2\mu\eta_0\alpha}{\alpha-1}\cdot\frac{T\alpha^{-t}-1}{\log_\alpha T}\right) &\le \frac{G^2\eta_0\alpha}{2\mu}\sum_{t=t^*+1}^{N}\frac{1}{\alpha^t}\exp\left(-\frac{\alpha^{t^*}}{\alpha^t} + \frac{2\mu\eta_0\alpha}{(\alpha-1)\log_\alpha T}\right) \\
&\le \frac{G^2\eta_0\exp\left(\frac{2\mu\eta_0\alpha}{(\alpha-1)\log_\alpha T}\right)}{2\mu}\cdot\frac{\alpha(1-2/\exp(1))}{\alpha^{t^*}} \\
&\le \frac{G^2(\alpha-1)\exp\left(\frac{2\mu\eta_0\alpha}{(\alpha-1)\log_\alpha T}\right)(1-2/\exp(1))}{4\mu^2}\cdot\frac{\log_\alpha T}{T},
\end{aligned}
$$
(33)

where the second inequality uses that $\int_{x=0}^{1} x\exp(-x)dx = 1 - 2/\exp(1)$. Incorporating (32) and (33) into (31) gives

$$\mathbb{E}\left[\left\|x_{S+1}^N - x^*\right\|^2\right] \le \frac{\left\|x_1^1 - x^*\right\|^2}{\exp\left(\frac{2\mu\eta_0\alpha}{\alpha-1}\cdot\frac{T-1}{\log_\alpha T}\right)} + \frac{G^2(\alpha-1)\exp\left(\frac{2\mu\eta_0\alpha}{(\alpha-1)\log_\alpha T}\right)}{4\mu^2}\cdot\frac{\log_\alpha T}{T}.$$

Changing the base $\alpha$ of $\log_\alpha$ to the natural logarithm, that is $\log_\alpha T = \ln T/\ln\alpha$, we arrive at the desired result. $\qquad\square$

Before proving the lower bound of Algorithm 2, we state an utility lemma.

**Lemma C.1.** *[24] Let $X_1, X_2, \cdots, X_K$ be independent random variables taking values uniformly from $\{-1, +1\}$ and $X = \frac{1}{K}\sum_{i=1}^{K}X_i$. Suppose $2 \le c \le \frac{\sqrt{K}}{2}$, then*

$$\mathbb{P}\left[X \ge \frac{c}{\sqrt{K}}\right] \ge \exp(-9c^2/2).$$

The lemma was proposed by [24] (see lemma 4) to show the tightness of the Chernoff bound. Recently it has been used to derive a high probability lower bound of the $1/t$ step-size [18]. We will now use Lemma C.1 to prove the following high probability lower bound of the step decay scheme in Algorithm 2 for $S = T/\log_\alpha T$.

*Proof.* (of **Theorem 4.2**) We consider the one-dimensional function $\tilde{f}_T(x) = \frac{1}{2}x^2$, where $x \in \mathcal{X} = [-4, 4]$. This function is 1-strongly convex and 1-smooth on $\mathcal{X}$. For any point $x_i^t$, the gradient oracle will return a gradient $x_i^t - z_i^t$ where $\mathbb{E}[z_i^t] = 0$. We apply the step decay step-size with $S = T/\log_\alpha T$ to $\tilde{f}_T$ using $x_1^1 = 0$ and $\eta_0 = 1$. Then the last iterate satisfies

$$x_{S+1}^N = \sum_{t=1}^{N} \sum_{i=1}^{S} \eta_t (1 - \eta_t)^{S-i} \Pi_{l>t}^N (1 - \eta_l)^S z_i^t. \tag{34}$$

Letting $t^* = \log_\alpha T - \log_\alpha \log_\alpha T + 1$, we have that $\eta_{t^*} = \eta_0/\alpha^{t^*-1} = \log_\alpha(T)/T$ and $(1-\eta_{t^*})^S = \exp(-1)$. For $t \neq t^*$ and $i \in [S]$, we pick $z_i^t = 0$. Then the final iterate $x_{S+1}^N$ can be estimated as

$$x_{S+1}^N = \eta_{t^*} \Pi_{l>t^*}^N (1 - \eta_l)^S \sum_{i=1}^{S} (1 - \eta_{t^*})^{S-i} z_i^t \geq \frac{\log_\alpha T}{\exp(2)T} \sum_{i=1}^{S} (1 - \eta_{t^*})^{S-i} z_i^{t^*}.$$

For $\nu_i^* = (1 - \eta_{t^*})^{S-i}$, it holds that $\exp(-1) < \nu_i^* < 1$ for all $i \in [S]$. Define $z_i^{t^*} = X_i^*/\nu_i^*$ where $X_i^*$ is uniformly chosen from $\{-1, +1\}$. Then $|z_i^{t^*}| \leq \exp(1) \in \mathcal{X}$ for any $i \in [S]$. Hence, this gradient oracle satisfies the assumptions. Now,

$$x_{S+1}^N \geq \frac{\log_\alpha T}{\exp(2)T} \sum_{i=1}^{S} X_i^* = \frac{1}{\exp(2)} \left( \frac{\log_\alpha T}{T} \sum_{i=1}^{S} X_i^* \right). \tag{35}$$

Invoking Lemma C.1 with $c = \sqrt{2\ln(1/\delta)}/3$ and $K = T/\log_\alpha T$ gives

$$f(x_{S+1}^N) = \frac{1}{2}(x_{S+1}^N)^2 \geq \frac{1}{2} \left( \frac{1}{\exp(2)} \frac{\sqrt{2\ln(1/\delta)}}{3\sqrt{T/\log_\alpha T}} \right)^2 = \frac{\ln(1/\delta)}{9\exp(2)\ln\alpha} \cdot \frac{\ln T}{T} \tag{36}$$

with probability at least $\delta > 0$. Since we know the optimal function value $\tilde{f}_T^* = \tilde{f}_T(x^*) = 0$, the the desired high probability lower bound follows. $\qquad\square$

*Proof.* (of **Theorem 4.3**) Recalling the iterate updates of Algorithm 2, for any $x \in \mathcal{X}$, we have

$$\mathbb{E}\left[\|x_{i+1}^t - x\|^2\right] = \mathbb{E}\left[\|\Pi_{\mathcal{X}}(x_i^t - \eta_t \hat{g}_i^t) - x\|^2\right] \leq \mathbb{E}\left[\|x_i^t - \eta_t \hat{g}_i^t - x\|^2\right]$$

$$\leq \mathbb{E}[\|x_i^t - x\|^2] - 2\eta_t \mathbb{E}[\langle g_i^t, x_i^t - x\rangle] + \eta_t^2 \mathbb{E}\left[\|g_i^t\|^2\right]$$

$$\leq \mathbb{E}[\|x_i^t - x\|^2] - 2\eta_t \mathbb{E}[\langle g_i^t, x_i^t - x\rangle] + \eta_t^2 G^2. \tag{37}$$

where the second inequality follows since $\|\Pi_{\mathcal{X}}(u) - v\| \leq \|u - v\|$ for any $v \in \mathcal{X}$ and the third inequality follows from the fact that the gradient oracle is bounded and unbiased. In the following analysis, we focus on the final phase, that is $t = N$. Let $k$ be the integer in $\{0, 1, 2, \cdots, S-1\}$. Extracting the inner product and summing over all $i$ from $S - k$ to $S$ gives

$$\sum_{i=S-k}^{S} \mathbb{E}\left[\langle g_i^N, x_i^N - x\rangle\right] \leq \frac{1}{\eta_N} \mathbb{E}\left[\|x_{S-k}^N - x\|^2\right] + (k+1)\eta_N G^2. \tag{38}$$

By the convexity of $f$ on $\mathcal{X}$, we have $\mathbb{E}[f(x) - f(x_i^N)] \geq \mathbb{E}\left[\langle g_i^N, x - x_i^N\rangle\right]$. Plugging this into (38), we get

$$\mathbb{E}\left[\frac{1}{k+1} \sum_{i=S-k}^{S} f(x_i^N) - f(x)\right] \leq \frac{1}{(k+1)\eta_N} \mathbb{E}\left[\|x_{S-k}^N - x\|^2\right] + \eta_N G^2. \tag{39}$$

We pick $x = x_{S-k}^N$ in (39) to find

$$\frac{1}{k+1} \sum_{i=S-k}^{S} \mathbb{E}[f(x_i^N)] - \mathbb{E}[f(x_{S-k}^N)] \leq \eta_N G^2. \tag{40}$$

Let $W_{k+1} = \frac{1}{k+1} \sum_{i=S-k}^{S} \mathbb{E}[f(x_i^N)]$ which is the average of the expected function values at the last $k+1$ iterations of the final phase $N$. The above inequality implies that

$$-f(x_{S-k}^N) \leq W_{k+1} + \eta_N G^2.$$

By the definition of $W_k$, we have $W_1 = \mathbb{E}[f(x_S^N)]$ and $kW_k = (k+1)W_{k+1} - \mathbb{E}[f(x_{S-k}^N)]$. Using these formulas and applying (40) gives

$$kW_k = (k+1)W_{k+1} - f(x_{S-k}^N) \leq (k+1)W_{k+1} - W_{k+1} + \eta_N G^2$$

which, after dividing by $k$, yields

$$W_k \leq W_{k+1} + \frac{\eta_N G^2}{k}.$$

Applying the above inequality recursively for $k = 1, \cdots, S-1$, we get

$$W_1 = \mathbb{E}[f(x_S^N)] \leq W_S + \eta_N G^2 \sum_{k=1}^{S-1} \frac{1}{k} \leq W_S + \eta_N G^2 (\ln(S-1) + 1). \tag{41}$$

It only remains to estimate $W_S$. At the $N^{\text{th}}$ phase, the iterate starts from $x_1^N$. We pick $x = x^*$ and $k = S-1$ in (39) so that

$$W_S := \mathbb{E}\left[\frac{1}{S} \sum_{i=1}^{S} f(x_i^N)\right] \leq f(x^*) + \frac{1}{\eta_N S} \mathbb{E}\left[\left\|x_1^N - x^*\right\|^2\right] + \eta_N G^2. \tag{42}$$

Note that in order to estimate $W_S$, we have to bound $\mathbb{E}\left[\left\|x_1^N - x^*\right\|^2\right]$ first. From inequality (30) of Theorem 4.1, we know that the distance between the starting point $x_1^N$ of the $N^{\text{th}}$ phase and $x^*$ can be bounded as follows:

$$\mathbb{E}\left[\left\|x_1^N - x^*\right\|^2\right] \leq \exp\left(-2\mu S \sum_{t=1}^{N-1} \eta_t\right) \left\|x_1^1 - x^*\right\|^2 + \frac{G^2}{2\mu} \sum_{t=1}^{N-1} \eta_t \exp\left(-2\mu S \sum_{l>t}^{N-1} \eta_l\right). \tag{43}$$

We now follow the proof of Theorem 4.1 to estimate $\mathbb{E}\left[\left\|x_1^N - x^*\right\|^2\right]$. Substituting the step-size $\eta_t = \eta_0/\alpha^{t-1}$ for $t \in [N]$, $N = \log_\alpha T$ and $S = T/\log_\alpha T$, we have

$$S \sum_{t=1}^{N-1} \eta_t = \frac{T}{\log_\alpha(T)} \frac{\eta_0(1 - \alpha^{-N+1})}{1 - 1/\alpha} = \frac{\eta_0 \alpha}{(\alpha-1)} \cdot \frac{T-\alpha}{\log_\alpha(T)}$$

$$S \sum_{l>t}^{N-1} \eta_l = \frac{T}{\log_\alpha(T)} \frac{\eta_0 \alpha^{-t}(1 - \alpha^{-(N-t-1)})}{(1 - 1/\alpha)} = \frac{\eta_0 \alpha}{\alpha-1} \cdot \frac{(T\alpha^{-t} - \alpha)}{\log_\alpha(T)}.$$

Therefore, using these inequalities in (43) gives

$$\mathbb{E}\left[\left\|x_1^N - x^*\right\|^2\right] \leq \exp\left(-\frac{2\eta_0 \mu \alpha}{\alpha-1} \cdot \frac{T-\alpha}{\log_\alpha T}\right) \left\|x_1^1 - x^*\right\|^2 + \frac{G^2(\alpha-1) \exp\left(\frac{2\mu \eta_0 \alpha^2}{(\alpha-1)\log_\alpha T}\right)}{4\mu^2} \cdot \frac{\log_\alpha T}{T}.$$

Incorporating the above results and substituting $\eta_N = \eta_0 \alpha/T$ and $S = T/\log_\alpha T$ into (41), we have

$$\mathbb{E}[f(x_S^N)] - f(x^*) \leq \frac{\mathbb{E}\left[\left\|x_1^N - x^*\right\|^2\right]}{\eta_N S} + \eta_N G^2 + \eta_N G^2(\ln(S-1) + 1)$$

$$\leq \frac{\left\|x_1^1 - x^*\right\|^2 \log_\alpha T}{\eta_0 \alpha \exp\left(\frac{2\eta_0 \mu \alpha}{\alpha-1} \cdot \frac{T-\alpha}{\log_\alpha T}\right)} + \frac{G^2(\alpha-1) \exp\left(\frac{2\mu \eta_0 \alpha^2}{(\alpha-1)\log_\alpha T}\right)}{4\mu^2 \eta_0 \alpha} \cdot \frac{\log_\alpha^2 T}{T}$$

$$+ \frac{G^2 \eta_0 \alpha(\ln(T) + 2)}{T}.$$

By changing the base of $\alpha$ to be natural logarithm, i.e., $\log_\alpha T = \ln T/\ln \alpha$, the proof is finished. $\square$

*Proof.* (of **Theorem 4.4**) Recall inequality (37) in the proof of Theorem 4.3: for any $x \in \mathcal{X}$ it holds that

$$\mathbb{E}[\|x_{i+1}^t - x\|^2] \leq \mathbb{E}[\|x_i^t - x\|^2] - 2\eta_t \mathbb{E}[\langle g_i^t, x_i^t - x\rangle] + \eta_t^2 G^2.$$

By the convexity of $f$, we have $\mathbb{E}[f(x) - f(x_i^t)] \geq \mathbb{E}[\langle g_i^t, x - x_i^t\rangle]$, so the above inequality implies that

$$2\eta_t \mathbb{E}[f(x_i^t) - f(x)] \leq \mathbb{E}[\|x_i^t - x\|^2] - \mathbb{E}[\|x_{i+1}^t - x\|^2] + \eta_t^2 G^2. \tag{44}$$

Let $t^* = \max\left\{0, \left\lfloor \log_\alpha\left(\frac{2\mu\eta_0\alpha^2}{\alpha-1} \cdot \frac{T}{\log_\alpha T}\right)\right\rfloor\right\}$ and $x = x^*$. By applying (44) repeatedly and summing over all $t^* \leq t \leq N$ and $i \in [S]$, we have

$$\sum_{t=t^*}^{N} \eta_t \sum_{i=1}^{S} f(x_i^t) - f(x^*) \leq \mathbb{E}\left[\|x_1^{t^*} - x^*\|^2\right] + SG^2 \sum_{t=t^*}^{N} \eta_t^2. \tag{45}$$

Let

$$\hat{x}_T := \frac{\sum_{t=t^*}^{N} \eta_t \sum_{i=1}^{S} x_i^t}{S \sum_{t=t^*}^{N} \eta_t}.$$

Since $\mathcal{X}$ is convex and each iterate $x_i^t$ belongs to $\mathcal{X}$, we have $\hat{x}_T \in \mathcal{X}$. By the convexity of $f$ and (45) it then follows that

$$\mathbb{E}[f(\hat{x}_T) - f(x^*)] = \mathbb{E}\left[\frac{\sum_{t=t^*}^{N} \eta_t \sum_{i=1}^{S} f(x_i^t)}{S \sum_{t=t^*}^{N} \eta_t}\right] - f(x^*) \leq \frac{\sum_{t=t^*}^{N} \eta_t \sum_{i=1}^{S} \mathbb{E}[f(x_i^t)]}{S \sum_{t=t^*}^{N} \eta_t} - f(x^*)$$

$$\leq \frac{\mathbb{E}\left[\|x_1^{t^*} - x^*\|^2\right]}{S \sum_{t=t^*}^{N} \eta_t} + \frac{SG^2 \sum_{t=t^*}^{N} \eta_t^2}{S \sum_{t=t^*}^{N} \eta_t}. \tag{46}$$

Next, we turn to estimate $\mathbb{E}\left[\|x_1^{t^*} - x^*\|^2\right]$. By (30) with $N = t^* - 1$, we have

$$\mathbb{E}\left[\|x_1^{t^*} - x^*\|^2\right] = \mathbb{E}\left[\|x_{S+1}^{t^*-1} - x^*\|^2\right] \leq \prod_{t=1}^{t^*-1}(1 - 2\mu\eta_t)^S \|x_1^1 - x^*\|^2 + \frac{G^2}{2\mu}\sum_{t=1}^{t^*-1}\eta_t \prod_{l>t}(1 - 2\mu\eta_l)^S$$

$$\leq \exp\left(-2\mu S \sum_{t=1}^{t^*-1}\eta_t\right)\|x_1^1 - x^*\|^2 + \frac{G^2}{2\mu}\sum_{t=1}^{t^*-1}\eta_t \exp\left(-2\mu S \sum_{l>t}^{t^*-1}\eta_l\right). \tag{47}$$

Next, we estimate the summation of $\eta_l$ from $l = t+1$ to $l = t^* - 1$

$$S\sum_{l>t}^{t^*-1}\eta_l = S \cdot \frac{\frac{\eta_0}{\alpha^t}(1 - (1/\alpha)^{t^*-t-1})}{(1 - 1/\alpha)} = \frac{T\eta_0}{\log_\alpha T}\left(\frac{\frac{1}{\alpha^{t-1}} - \frac{1}{\alpha^{t^*-2}}}{\alpha - 1}\right) = \frac{\eta_0}{\alpha - 1} \cdot \frac{T}{\log_\alpha T \alpha^{t-1}} - \frac{1}{2\mu}. \tag{48}$$

Incorporating (48) into the second term of (47) gives

$$\sum_{t=1}^{t^*-1}\eta_t \exp\left(-2\mu S\sum_{l>t}^{t^*-1}\eta_l\right) = \eta_0\sum_{t=1}^{t^*-1}\frac{1}{\alpha^{t-1}}\exp\left(-\frac{2\mu\eta_0}{(\alpha-1)\alpha^{t-1}} \cdot \frac{T}{\log_\alpha T} + 1\right)$$

$$= \frac{\eta_0\alpha^2 \exp(1)}{\alpha^{t^*}}\sum_{t=1}^{t^*-1}\frac{\alpha^{t^*}}{\alpha^{t+1}}\exp\left(-\frac{\alpha^{t^*}}{\alpha^{t+1}}\right)$$

$$\leq \frac{2\eta_0\alpha^2}{\alpha^{t^*}} = \frac{(\alpha-1)}{\mu} \cdot \frac{\log_\alpha T}{T}, \tag{49}$$

where the inequality follows from the fact that $\int_{x=1}^{+\infty} x\exp(-x)dx \leq 2/\exp(1)$. Letting $t = 0$ in (48), we have

$$S\sum_{l=1}^{t^*-1}\eta_l == \frac{\eta_0\alpha}{\alpha - 1} \cdot \frac{T}{\log_\alpha T} - \frac{1}{2\mu}. \tag{50}$$

Plugging (49) and (50) into (47), we get

$$\mathbb{E}\left[\left\|x_1^{t^*} - x^*\right\|^2\right] \leq \frac{\left\|x_1^1 - x^*\right\|^2}{\exp\left(\frac{2\mu\eta_0\alpha}{\alpha-1} \cdot \frac{T}{\log_\alpha T} - 1\right)} + \frac{G^2(\alpha-1)}{2\mu^2} \cdot \frac{\log_\alpha T}{T}. \tag{51}$$

Incorporating (51) into (46) and using $\frac{\alpha-1}{2\mu\alpha} \leq S\sum_{t=t^*}^N \eta_t \leq \frac{1}{2\mu}$ and $S\sum_{t=t^*}^N \eta_t^2 \leq \frac{\log_\alpha T}{4\mu^2(\alpha+1)T}$ gives

$$\begin{aligned}
\mathbb{E}[f(\hat{x}_T) - f(x^*)] &\leq \frac{\mathbb{E}[\left\|x_1^{t^*} - x^*\right\|^2]}{S\sum_{t=t^*}^N \eta_t} + \frac{SG^2 \sum_{t=t^*}^N \eta_t^2}{S\sum_{t=t^*}^N \eta_t} \\
&\leq \frac{2\mu\alpha\mathbb{E}[\left\|x_1^{t^*} - x^*\right\|^2]}{\alpha-1} + \frac{\alpha G^2}{2\mu(\alpha^2-1)} \cdot \frac{\log_\alpha T}{T} \\
&\leq \frac{2\mu\alpha}{\alpha-1} \cdot \frac{\left\|x_1^1 - x^*\right\|^2}{\exp\left(\frac{2\mu\eta_0\alpha}{\alpha-1} \cdot \frac{T}{\log_\alpha T} - 1\right)} + \frac{\alpha\left(1 + \frac{1}{2(\alpha^2-1)}\right)G^2}{\mu} \cdot \frac{\log_\alpha T}{T}
\end{aligned}$$

which concludes the proof. $\qquad\square$

## D. The Details of the Setup in Numerical Experiments

In this section, we provide some details for the numerical experiments in Section 5 and give some complementary experimental results.

To better understand the relationship between all the considered step-sizes, we draw Figure 5 to show the step-size $\eta_t$ ($y$-axis is $\log(\eta_t)$) versus the number of iterations (starting from the same initial step-size). In the left picture, we show the many step-sizes, studied in Sections 5.1 and 5.2, which finally reach the order of $1/\sqrt{T}$ for the non-convex and convex cases. In the strongly convex case (the right picture), we show the step-sizes which are based on the order of $1/T$. We also add yet another kind of exponentially decaying step-size (called Exp(H-K-2014)) proposed by [19]: $\eta_i = \eta_0/2^i$, $t \in [T_i, T_{i+1})$ and $T_{i+1} = 2T_i$, where $\sum_i T_i = T$.

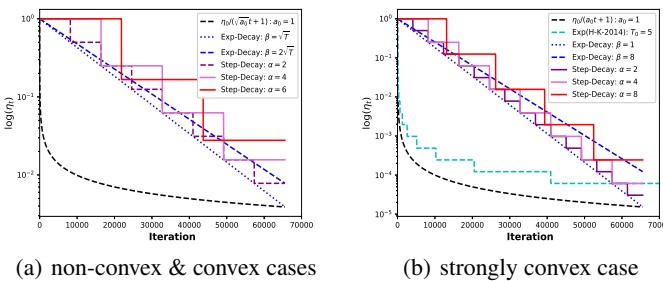

(a) non-convex & convex cases      (b) strongly convex case

Figure 5: Step-sizes involved in the experiments

From Figure 5(a), we observe that Exp-Decay for $\beta = \sqrt{T}$ can be regarded as a lower bound of Step-Decay. From another viewpoint, we can see that when the decay factor $\alpha$ is very close to 1, the proposed Step-Decay will reduce to Exp-Decay for $\beta = \sqrt{T}$. A similar relationship can also be observed from Figure 5(b).

### D.1 The Details of the Experiments on MNIST

The MNIST dataset consists of a training set of 60,000 examples and a testing set of 10,000 examples. We train MNIST on a fully connected two-layer network (784-100-10). The $l_2$ regularization parameter is $10^{-4}$ and the mini-batch size is 128. We run 128 epochs, which implies that the number of iterations $T$ is equal to the training size $(60,000)$. The experiments on MNIST are implemented in Python 3.8.5 on the server with two 3.1 GHz Intel Xeon processors (total 32 cores) and 128 GB of RAM.

In order to fairly compare the considered step-sizes, the initial step-size $\eta_0$ is chosen from the search grid $\{0.001, 0.005, 0.01, 0.05, 0.1, 0.5, 1, 5\}$. For $1/t$ and $1/\sqrt{t}$, the initial step-size is $\eta_0 = 1$, and $a_0$ is tuned by searching for the final step-size $\eta_T$ over the grid $\{0.001, 0.005, 0.01, 0.05, 0.1, 0.5, 1, 5\}$; in our experiments, it turned out that the best value of $\eta_T$ was $0.01$. The best initial step-size $\eta_0$ was found to be $0.5$ for both Exp-Decay and Step-Decay. Similarly, the parameter $\beta$ for Exp-Decay is selected to make sure that its final step-size $\eta_T$ is tuned over the grid $\{0.001, 0.005, 0.01, 0.05, 0.1, 0.5, 1, 5\}$; the best tuning of $\eta_T$ was found to be $0.05$. For Step-Decay, the decay factor $\alpha$ is empirically chosen from an interval $(1, 12]$ and the search grid is in units of $1$ after $\alpha \geq 2$. The outer-loop size $N$ is $\lfloor \log_\alpha T/2 \rfloor$ which is numerically better than its ceil. The best choice was found to be $\alpha = 7$ ($N = 2$).

### D.2 The Details of the Experiments on CIFAR10 and CIFAR100

The benchmark datasets CIFAR10 and CIFAR100 both consist of 60000 colour images (50000 training images and the rest 10000 images for testing). The maximum epochs called for the two datasets is 164 and batch size is 128. The numerical experiments for CIFAR10 and CIFAR100 are implemented in Python 3.7.4 with 2 x Nvidia Tesla V100 SXM2 GPU with 32GB RAM.

First, we employ a 20-layer Resident Network model [21] called ResNet20 to train CIFAR10. We use vanilla SGD without dampening and a weight-decay of $0.0005$. The hyper-parameters are selected to work best according to their performance on the test dataset. For all evaluated step-sizes, the initial step-size $\eta_0$ is tuned from $\{0.0001, 0.0005, 0.001, 0.005, 0.01, 0.05, 0.1, 0.5, 1\}$. For the constant step-size, the best choice was achieved by $\eta_t = 0.05$. For the $1/t$ step-size, the initial step-size $\eta_0 = 1$ and the parameter $a_0$ is tuned such that $\eta_T$ reaches the search grid $\{0.0001, 0.0005, 0.001, 0.05, 0.01, 0.5, 0.1\}$ (the grid search yielded $\eta_T = 0.05$). We tuned the $1/\sqrt{t}$ step-size in the same way as the $1/t$ step-size: the initial step-size $\eta_0 = 1$ and $\eta_T = 0.05$. For Exp-Decay, $\eta_0 = 1$ and $\beta$ is chosen such that the final step-size $\eta_T$ reaches the search grid for step-size (resulting in $\eta_T = 0.01$). For Step-Decay, the best initial step-size is achieved at $\eta_0 = 0.5$ and $\alpha = 6$.

The numerical results on CIFAR10 is shown in Figure 6. We can see that the sudden jumps in step-size helps the algorithm to get a lower testing loss and higher accuracy (red curve) compared to other step-sizes.

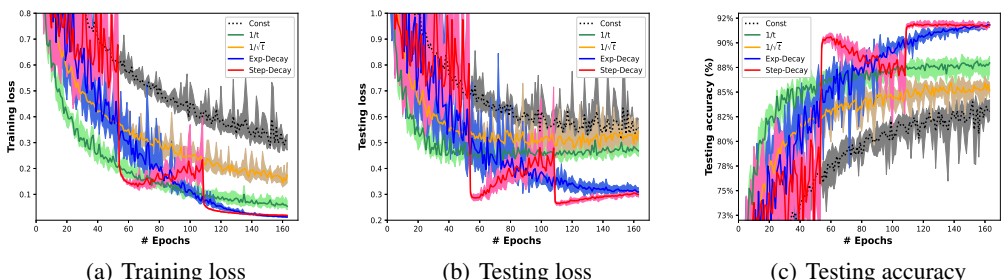

| (a) Training loss | (b) Testing loss | (c) Testing accuracy |

Figure 6: Results on CIFAR10 - ResNet20

Next, we give the details about how to select the optimal values of the parameter $\eta_0$, $a_0$, $\beta$ and $\alpha$ for CIAFR100. The initial step-size for all step-sizes is chosen from $\{0.0001, 0.0005, 0.001, 0.005, 0.01, 0.05, 0.1, 0.5, 1\}$. For the constant step-size, we found $\eta_0 = 0.1$ and a weight-decay of $0.0001$. For the $1/t$ step-size, $\eta_0 = 1$ and $a_0$ was set such that $\eta_T$ is searched over the grid above, which yielded $\eta_T = 0.01$. For $1/\sqrt{t}$, $\eta_0 = 1$ and $a_0$ is set to make sure that $\eta_T$ is tuned from the set for step-size, resulting in $\eta_T = 0.01$. For Exp-Decay, $\eta_0 = 1$ and $\beta$ is chosen such that $\eta_T$ reaches the grid $\{0.0001, 0.0005, 0.001, 0.05, 0.01, 0.5, 0.1\}$ (resulting in $\eta_T = 0.01$). For the Step-Decay, the initial step-size $\eta_0 = 1$ and the decay factor $\alpha = 6$.

Next, we detail the parameter tuning for the algorithms in Table 1. The maximum epochs called was 164 and the batch size was 128. This implies that $T = 164 \cdot T/128$. The weight-decay was set to 0.0005 for Adam and NAG. The best initial step-size for AdaGrad was found to be $\eta_0 = 0.05$ (weight-decay is 0); For Adam, we found the parameters $(\beta_1, \beta_2) = (0.9, 0.99)$. The weight-decay

was set to 0.025 for AdamW while the other parameters were the same as for Adam. For NAG, the momentum parameter was set to 0.9. For Exp-Decay: the best-tuned $\beta$ was $0.005 \cdot T$ and $\eta_0 = 0.1$ for NAG; $\beta = 0.01 \cdot T$ and $\eta_0 = 0.005$ for Adam and AdamW; For Step-Decay: the optimal $\alpha$ was found to be 6 for all methods including Adam, AdamW and NAG while the best $\eta_0 = 0.05$ for NAG and $\eta_0 = 0.005$ for Adam and AdamW.

In Figure 7(a), we show how the number of outer-loop iterations $N$ changes with the decay factor $\alpha \in (1, 12]$. The decay factor is an important hyper-parameter for Step-Decay. To figure out the decay factor affects the performance, we plot the testing loss and generalization error (the absolute value of the difference between training loss and testing loss), as well as the testing accuracy in Figures 7(b) and 7(c), respectively. All results are repeated 5 times. The best choice of the decay factor is found to be $\alpha = 6$, according to the best performance on testing loss and accuracy. It is observed that $\alpha \in [4, 6]$ performs better and is more stable than $\alpha \in [7, 12]$. The main reason is that the length of each phase for $\alpha \in [7, 12]$ is larger than that of $\alpha \in [4, 6]$ so that it loses its advantages in the end (the generalization is weakened). Moreover, suppose that the number of outer-loop iterations $N$ is fixed, for example at $N = 3$ (where $\alpha \in [4, 6]$) or $N = 2$ (where $\alpha \in [7, 12]$), we can see that the testing loss is getting better if we increase the decay factor.

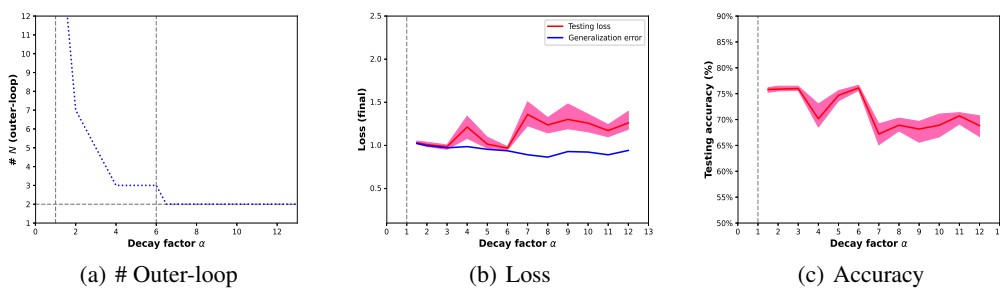

(a) # Outer-loop        (b) Loss        (c) Accuracy

Figure 7: The performance of decay factor $\alpha$ on CIFAR100

### D.3 Numerical Details for Regularized Logistic Regression

In this part, we present the numerical results for all considered step-sizes on regularized logistic regression. The initial step-size $\eta_0$ is best-tuned from the search grid $\{0.01, 0.05, 0.1, 0.5, 1, 10, 50, 100\}$ for all step-sizes. For the constant step-size, the initial step-size is $\eta = 1$. For $1/t$, $1/\sqrt{t}$, Exp(H-K-2014), Exp-Decay [28] and Step-Decay, the initial step-size $\eta_0$ is 10. We tune $a_0$ for $1/t$ and $1/\sqrt{t}$ step-sizes such that the final step-size $\eta_T$ is searched over the grid $\{0.01, 0.05, 0.1, 0.5, 1, 10, 25, 50, 75, 100\}$ ($\eta_T = 0.1$). Similarly, the parameter $\beta$ of Exp-Decay [28] is chosen such that $\eta_T$ is searched over the grid $\{0.01, 0.05, 0.1, 0.5, 1, 10, 50, 100\}$ ($\eta_T = 0.01$). The initial period $T_0$ for Exp(H-K-2014) is $T_0 = 5$. For Step-Decay, the decay factor is chosen to be $\alpha = 4$.

Compared to the polynomial diminishing step-sizes (e.g. $1/t$, $1/\sqrt{t}$), we can observe that Exp-Decay and Step-Decay not only yields rapid improvements initially, but they also converge to a good solution in the end.

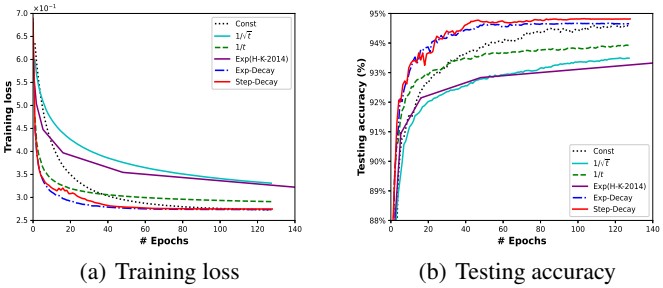

(a) Training loss        (b) Testing accuracy

Figure 8: The results on rcv1.binary - logistic($L_2$)