# OpenReview forum: "On the Convergence of Step Decay Step-Size for Stochastic Optimization"
_NeurIPS.cc/2021/Conference — NeurIPS 2021 Poster_

### Official Review · Reviewer_Dkst · 2021-07-14

**Rating:** 7
**Confidence:** 3

**Summary:**

This paper proved convergence results for SGD with step decay step-size (the step size is decayed by some factor after certain iterations).

In the smooth non-convex setting, this paper sampled the output from $\\{x_t\\}$ with probability of $x_t$ being $O(1/\eta_t).$ Based on this novel approach of sampling the output, the authors
1. proved the first convergence bound for SGD with step-decay step-sizes on non-convex problems and achieved a $O(\ln T/\sqrt{T})$ rate towards a stationary point, which is comparable to the folklore results for $\eta_t = O(1/\sqrt{t})$ step-size;
2. improved the convergence results for exponentially decaying step-size, which allows larger initial step size compared with previous work;
3. improved the convergence rate from $O(\ln T/\sqrt{T})$ to $O(1/\sqrt{T})$ for the $1/\sqrt{t}$ step-size.

For the strongly convex and smooth functions, this paper proved an $O(\ln T/T)$ convergence rate for the last iteration, which generalized the result of Ge et al. The paper also proved a $O(\ln^2 T/T)$ convergence rate for the strongly convex and non-smooth functions.

In the experiments, the authors verified that step-decay step-size scheduling have better performance in training neural networks compared with constant step size or polynomially decaying step-size.

**Limitations And Societal Impact:**

The authors have adequately addressed the limitations and potential negative societal impact of their work.

**Main Review:**

I think this is a good theory paper that studied the convergence rate of SGD for different class of functions under step decay step-size. There are not many theoretical guarantees for SGD with step decay step-size. I believe this paper is a good step along this direction.

It's an interesting idea to sample the output from the trajectory with the probability of each point inverse to the step size. This novel sampling scheme allowed the proof of the convergence bounds with step-decay step-sizes and also allowed the improvement of some other results. I think this sampling scheme might be of interests also in other settings, and also might be useful in practice.

Some suggestions:
Since this is mostly a theory paper, I think it might be good to talk more about the challenges in the proof and also the technical contributions. Currently, the results are well explained, but it's unclear what the novelty of the analysis is and how the proof differs from the previous works.

**Time Spent Reviewing:**

4

---

> ### Author Response · Authors · 2021-08-10
> **Response to Reviewer Dkst**
>
> Thank you for your positive review and constructive comments.
>
> We clarify our theoretical novelty in Section 3.1 before Theorem 3.2, but we agree that we could have extended this discussion. As you proposed, we will highlight the theoretical significance and give more details to show the differences compared with the proofs in previous works in the final version of our paper. Thanks.

---

> > ### Comment · Reviewer_Dkst · 2021-08-28
> > **Thanks for the response**
> >
> > Thanks for the response. After reading the response and other reviews, I keep my current positive review.

---

### Official Review · Reviewer_5Mza · 2021-07-16

**Rating:** 7
**Confidence:** 4

**Summary:**

The authors present a new analysis of SGD with the very popular step decay implemented in all DL framework for instance. They provide convergence guarantee in various setting and show that the classical exponential decay is a very special case of their result, still achieving the state of the art result in this special case. They also provide some lower bound to show tightness of their results and finally propose a new sampling of the iterates that allows to remove some annoying log term in the convergence bound.

**Limitations And Societal Impact:**

I think this section does not apply for this work.

**Main Review:**

\paragraph{Strengths:}

The authors provide new convergence result on a very popular setting, used in practice but not studied theoretically. They provide some comprehension on this scheduler and their general bound allows to understand how to choose optimally the length of the plateau.
Moreover their general analysis is sufficiently tight to be able to recover known results of another scheduler as a special case.

\paragraph{Weaknesses:}

In Theorem 4.1, 4.3 and 4.4, the condition "bounded gradient oracle" with the strong convexity is very problematic in my humble opinion. Indeed, bounded gradient oracle implies that the (deterministic) gradient of the objective function is bounded as well, while strong convexity implies it is lower bounded by the distance of the iterate to optimum. The 2 conditions combined together implies that the authors assume the iterates to be bounded, which is problematic to me as this is not an assumption on the problem, but on the result itself, excluding some possible scenario.
A minor weakness is that to reach the sqrt convergence bound, one needs to know in advance the number of iterations.

\paragraph{Correctness:}

I have not check all the proofs in details, but from what I checked, I have not spotted any mistake.

\paragraph{Clarity:}

The claim is clear and results follow. I would still make a couple of very minor suggestions to make some point clearer:

- Definition 4: I m a bit confused by the definition of subgradient. I generally consider the subgradient is precisely the set of vector verifying the inequality provided in Definition. Therefore a convex function is a function which subgradient is non empty in any point. And for non convex function, we need a new definition of the subgradient. For sake of clarity, could the authors add the definition of $\partial f(x)$ before using it?
- Theorem 3.2 l.166: it became clear to me after equation 5, that $N = \log_{\alpha} T / 2$ actually meant $\frac{\log_{\alpha} T}{2}$, and not $\log_{\alpha} \frac{T}{2}$. In my humble opinion, the lack of parenthesis is confusing. Could the authors either add parenthesis, or use "frac" or replace the division by 2 by a sqrt inside the log? I think it would improve clarity here.
- l.181: I think the authors means that when $V$ gets large, it doesn't slow too much the convergence of the algorithm in the sense we can bound the rate uniformly in $V$. In that sense, I think we can say, the convergence bound inn not affected by very large $V$. However, "the convergence bound unaffected by the noise variance" can bring a confusion as a reader can understand that the bound does not depend at all of $V$ and we can derive, in the stochastic setting, the deterministic setting's bound, which I think is not the case here.

\paragraph{General remarks and suggestions:}

\begin{itemize}
- l.16: "can either be"
- l.67: "smooth with Lipschitz continuous gradient": isn't repetitive?
- Algorithm 1: To my point of view, " inner-loop size S = T/N" should go after the input line, as $S$ is not an input but some value the method computes from inputs.
- l.175: there is a typo here, a $T$ is missing into the log
- l.191-192: the authors propose here to consider $\alpha = \left(\frac{\beta}{T}\right)^{-1/T}$ and then $\beta = O(\sqrt(T))$. Note that Theorem 3.2 suggests to consider $N$ such that $\alpha^N = \sqrt(T)$. In the special case where $S=1$, then $N=T$, this equality suggests the choice of $\alpha$ to be $alpha = \sqrt(T)^{1/T}$ which is aligned with l.191-192's choice. I think it would worth adding a word on this. Especially since the authors say that the choice $\alpha = \left(\frac{\beta}{T}\right)^{-1/T}$ comes from another paper. I think they should stress they not only make this choice based on another paper, but that it is also a consequence of a very special case of their own Theorem.
\end{itemize}

\paragraph{Conclusion}

I think this paper brings interesting new analysis of some very used technique in practice. In my humble opinion, this type of results is of great importance for the community to understand what makes this scheduler works well and then have insight on how to tune it (choose N or equivalently S, and choose alpha). Even if the proposed tuning depends on inaccessible noise of gradient, or on the final number of iterations, I think those results are interesting and open future work. For instance estimate V to try to set $\alpha$ without exterior knowledge, or set $N$ independently of $T$ by for instance, changing it online. I would vote the accept the paper. However, in my humble opinion authors should consider either removing theorems using contradictory assumptions or fix them if possible.

**Time Spent Reviewing:**

4-5

---

> ### Author Response · Authors · 2021-08-10
> **Response to Reviewer 5Mza**
>
> We are grateful for your positive feedback and very insightful suggestions.
>
> **Regarding the assumption of bounded stochastic gradient on strongly convex case and prior knowledge on $T$:**
>
> Thank you. In Theorems 4.1, 4.3 and 4.4, we consider constrained and possibly non-differentiable problems and can thus use the bounded-gradient assumption without loss of generality (see, e.g., Rakhlin et al., 2012; Shamir & Zhang, 2013). As future work, it will be interesting to explore how to relax the assumption to avoid the possible drawbacks on the convex constrained setting. In the unconstrained case it is straightforward to relax this assumption to bounded-variance or additive multiplicative noise model.
>
> We agree that we have to assume prior knowledge about the total number of iterations $T$. We are currently looking into adaptive schemes where $\alpha$ is fixed, $S$ is adapted so that the iterates reach stationarity, and $T$ is free, but have not yet obtained any sharp theoretical guarantees.
>
>
> **Regarding the definition of subgradient:**
>
> Thank you for pointing this out. We are sorry for the misunderstanding on this point. We will add the definition of the subgradient in the final version. For the convex and strongly convex cases, we consider  nonsmooth and constrained problems, so we only get access  to the subgradient. In the nonconvex case, we consider the constraint $X = R^d$ and $L$-smooth functions (Lipischitz continuous gradient). So in this case, we have access to the gradient. We will clarify this in the final version of the paper..
>
> **Regarding the notation of $\log_{\alpha}T/2$:**
>
> We are sorry for the confusion on this notation. We will modify this to $(\log_{\alpha} T)/2$ in the final version.
>
> **Regarding the line 181:**
>
> Thanks for your comments. We agree that when the noise level $V$ gets larger, the convergence bound is not affected by very large $V$. We will modify our statement afterwards.
>
> **Regarding General remarks and suggestions:**
>
> Thank you for pointing these out. We will fix lines 16, 67, 175, and Algorithm 1 as you suggest.
>
> Your comments on lines 191-192 are valuable suggestions. Thanks! As you proposed, we will clarify and stress the connections of Theorem 3.2 and Theorem 3.3 in the final version of our paper.
>
> Finally, we are grateful for the reviewer to share some interesting and open questions for future work. We will also try our best to make this paper even clear and explore how to relax the assumption to avoid the possible drawbacks in the future.

---

### Official Review · Reviewer_AbDY · 2021-07-16

**Rating:** 7
**Confidence:** 3

**Summary:**


This paper studies theoretical properties and practical efficiencies of a step decay step-size policy for non-convex, convex and deep learning problems. The main contribution is to establish a novel convergence guarantee for a practical step decay step-size scheme of SGD beyond existing results on least-squares problems.


**Ethics Review Area:**

["I don’t know"]

**Limitations And Societal Impact:**

yes

**Main Review:**


It is known that decaying step sizes too fast may result in poor performance in SGD. Step decay step-size policies are hence often used in non-convex optimization such as deep learning problems. They can sometimes achieve a fast convergence towards solutions with a good generalization performance. This papers focuses on a practical step decay step-size policy and two stopping rules (either a random iterate or the last iterate), and analyzed their theoretical properties. The practical performance are then evaluated against popular step-size policies.

The paper is well-organized and well-written.
I have a few questions/comments:
* The first main contribution, such as the removal of ln(T) factor for the 1/sqrt(t) step-size, comes from an important assumption that the objective function is upper bounded, isn’t it? It seems that this assumption is in all the results of non-convex cases, therefore it is better to be mentioned in the introduction, e.g. to make a comparision on such assumptions with related works.
* In Figure 3, why the method with Step-decay has a loss that is increased over epochs 60-110?
* Enlarge the legend size in all the figures if possible. They are too small to read.
* Typo: line 16 can be, line 77 for.


**Time Spent Reviewing:**

7

---

> ### Author Response · Authors · 2021-08-10
> **Response to Reviewer AbDY**
>
> We are grateful for your  positive feedback and thoughtful comments.
>
> **Response to Q1 about $1/\sqrt{t}$ step-size and the assumption:**
>
> Not really. As shown in Proposition 3.1, if we use the probability sampling rule $P_t  \propto \eta_t$, we can only get the near-optimal (up to $\ln T$) for $1/\sqrt{t}$ step-size. With this output probability distribution, the assumption that the objective function is bounded is not useful. By using the probability sampling rule $P_t  \propto  1/\eta_t$ for the output (that we propose), we can remove the $\ln(T)$ term for $1/\sqrt{t}$ step-size. But to derive this rate we assume that the objective function is bounded.  We should also highlight that the assumption that the objective function should be bounded can be relaxed, see our answer to Q1 of reviewer JpYA.   Regarding whether the assumption is only needed for the non-convex case, then the answer is yes, we don’t need it in the convex cases.  We appreciate the suggestion to discuss the assumptions in the introduction and to make a better comparison of the assumptions used in the literature. We will definitely do so  in the final version.
>
> **Response to Q2 about the loss increased:**
>
> Thank you for your comments. We discuss this phenomenon a little bit in the conclusion.  In the experiments, we observed that as the iterates proceed in each stage (especially in the second stage), their performance gets worse. A similar phenomenon is also observed in references [28, 48]. Our guess is that the noise accumulates quickly under the long terms of constant step-size. Therefore, it will be interesting to study how to best select the size of inner-loop $S$ (instead of constant or exponentially growing) to decide when to drop the step-size and avoid the loss of generalization.
>
> **Response to Q3 about the small size in the figures:**
>
> Thanks. We agree the font size should be increased, we will fix it in the final version.
>
> **Response to Q4 about the small typos:**
>
> Thank you for pointing out these typos. We will fix them in the final version.

---

### Official Review · Reviewer_JpYA · 2021-07-19

**Rating:** 5
**Confidence:** 4

**Summary:**

This paper investigates the step decay schedule (constant then cut) for SGD. It provides theoretical analyzes of this strategy in strongly-convex, convex, and non-convex settings showing near-optimal results. Empirical experiments illustrate its advantage over other popular optimizers.

**Limitations And Societal Impact:**

Yes

**Main Review:**

The step decay (constant then cut) strategy is widely used in training deep neural networks. There have been quite a few works dedicated to it, both theoretical and empirical. This paper does a great job in synthesizing the related works, identifying their contributions and limits, and pointing out where this work stands.

This strategy is of course not novel, nor are the rates they obtained in any setting, but providing the theoretical guarantees for the step decay schedule as proved in this paper for various settings are new. In addition, the sampling scheme of being inversely proportional to $\eta_t$ is novel and very intuitive in giving weights to last iterates.

Some of my concerns are:
1. The assumption on the upper-boundedness of $f$ is worrisome, especially when it appears in the bound with the term $f_{max} - f^*$ showing that if it is unbounded then the bound is trivial. If the domain is bounded then this assumption would be more natural but for unbounded domain, this assumption seems too restrictive. Note that other results typically include a term $f(x_1) - f^*$ which is fine as $f(x_1)$ can be manually selected and easily be bounded. Looking at the proofs, however, the term $f_{max} - f^*$ is not just a simple upper-bound of $f(x_1) - f^*$ and the assumption on $f_{max} < \infty$ is indispensable.
2. In the paper on exponential step sizes [Li et al., 2020], they also studied the cosine step sizes and showed that it also performs really well and surpasses the exponential step sizes and the step decay in some settings. Also, in their paper, the step decay schedule has the flexibility of using different inner-loop sizes thus should be different from your definition in this paper. Hence, I would like to see a comparison with it.
3. I see that you answered Yes to Checklist 3.a, but could not find your codes on implementing the algorithm and comparing with other optimizers, apart from download link to datasets you used.

**Time Spent Reviewing:**

6

---

> ### Author Response · Authors · 2021-08-10
> **Response to Reviewer JpYA**
>
> We are grateful for the positive feedback and constructive comments. Your comments and concerns are addressed below one by one.
>
> **Regarding Q1 about the upper boundedness of function f:**
>
> First note that we only need this assumption for nonconvex functions, not for convex functions. Second, the actual assumption we need for our results to hold in the non-convex case is much more relaxed. In particular, we only need $E[f(x_t) - f*] $ to be bounded for every $t$. If the objective function is continuous, then this assumption holds as long as the iterates $x_t$ are bounded. Boundedness of iterates is a common assumption in the stochastic approximation literature (see e.g. Ljung (1977), Kushner and Yin (1997), Borkar (2008)) and in many papers on stochastic optimization (e.g. https://arxiv.org/pdf/1804.07795.pdf by Davis et al). It is only very recently that such boundedness results have been proven for specific step-sizes using basic principles (e.g. https://arxiv.org/abs/2006.11144). We believe that it is a fair assumption to make, and it has never been violated in our numerical experiments.
>
> Another way to achieve this assumption is to assume that the objective function is globally bounded, as we did in the initial version of the paper. As the reviewer points out this might be a bit of a restrictive assumption, but still it holds in many interesting cases. For example, truncated loss functions under heavy-tailed distributions satisfy this assumption [e.g., http://proceedings.mlr.press/v115/xu20b.html from Xu et. al. 2020]. The truncated technique could be used by all standard loss functions. This function is more robust to outliers and has been adopted by practitioners [Belagianni et al., 2015] and widely used for robust machine learning [Park & Liu, 2011,  Catoni, 2012, Brownlees et al. 2015].   Moreover, this assumption is a standard assumption in statistical learning theory [Vapnik 1998, 2006; Cortes et al., 2019] for the generalization bound.
>
> Note that this assumption does not imply that the random loss function $f(x; \xi)$ for $\xi \in \Xi$ is bounded, but only that it has a bounded mean (the mean is the the objective function in our case), which can be derived (or implied) from the assumptions in some recent papers [Xu et. al. 2020;  Cortes et al., 2019 https://link.springer.com/content/pdf/10.1007/s10472-018-9613-y.pdf].
>
> **Regarding Q2 about the paper  [Li et al., 2020]:**
>
> As the reviewer proposed more detailed comparisons with the related paper [Li et al., 2020], we clarify the main differences with [Li et al., 2020] below. The paper has been published in ICML 2021, we will update the citation as [Li et al., 2021].
>
> * We agree that cosine step-size performs well in some settings from the recent paper [Li et al, 2021]. We focus on the step-decay step-size and exponential decay step-size and do not give any analysis on the cosine step-size, so we did not implement this step-size in our experiments. But it will be interesting to see its performance and to check if our analysis can immediately apply to the cosine step-sizes.
> * In  Li’s paper, they focus on the theoretical convergence about cosine step-size and exponential step-size. The exponential step-size is a special case of step-decay step-size (when the inner loop size is 1). We discuss this in Section 3.2, where we also improve the convergence guarantees for the exponential step-size compared to those in [Li et al., 2021].
> * In  Li’s paper, they did not provide a theoretical analysis for the step-decay step-size as we do. But in the numerical experiments, they choose one milestone or two milestones step-decay step-size (that is outer-stage length $N = 2$ or 3, decay factor $\alpha$ =10) as the baseline of their numerical experiments. In our analysis, we also provide theoretical guarantees for such step-decay step-size with free parameter $N$ and decay factor $\alpha$ (see the Inequality (4) in Theorem 3.2). To achieve a near-optimal rate, we embody the relationship of $N = (\log_{\alpha}T)/2$. Our numerical experiments also verify the effectiveness of this choice.
>
> **Regarding Q3 about the code:**
>
> Thanks for your comments. We uploaded our code to the anonymous github system, please click this link https://anonymous.4open.science/r/Step-Decay-Step-Size-8724 for more details (this system is double blind without any identifying information). In the paper, we also gave the details about the settings of all the experiments and the important parameters in Section 5 and Appendix D. The reader should be able to reproduce the main numerical results using the details we provide.

---

> > ### Comment · Reviewer_JpYA · 2021-09-02
> > **Thanks for the response**
> >
> > After reading your response and other reviews, I have decided to maintain the score.
> >
> > First, I don't see how that $E[f(x_t) - f^*]$ needs to be bounded for every $t$ is much relaxed than the upper-boundedness of $f$, so I would still take that you basically require $f$ to be upper-bounded. Otherwise, the term $\Delta_0 = f_{max} - f^*$ would be unbounded and render the bound meaningless.
> >
> > Second, the reason I am citing cosine step sizes and step decay with arbitrary milestones is to check your algorithm's performance compared with the SOTA. However, this comparison has not been provided so I remain concerned.

---

> > > ### Author Response · Authors · 2021-09-02
> > > **Response to Reviewer JpYA**
> > >
> > > Dear reviewer - thank you for letting us know about your remaining concerns.
> > >
> > > Actually, it is a misconception that $f$ has to be bounded. On the contrary, a more useful class of problems are those where the loss function tends to infinity as the norm of $x$ tends to infinity (for example, regularized training problems). Intuitively, in such problems, the gradient will eventually grow sufficiently large to hold the iterates $x_t$ back from growing. A more formal argument becomes more technical, but you can for example see the paper https://arxiv.org/pdf/2004.06977.pdf (not written by us) for related results.
> > >
> > > Second, if you compare Figure 6 in the supplementary to our submission with Table 1 in Li et al 2021 http://proceedings.mlr.press/v139/li21z/li21z.pdf, you will see that our results are comparable to stagewise -2 milestones, exponential step-size and cosine step-size. We attain a testing accuracy for CIFAR10, which is even slightly better than stagewise -2 milestones provided in Li et al 2021. It is difficult to compare results on other datasets, since the experimental settings are different.
> > >
> > > Thank you very much for providing valuable comments on our paper.

---

### Author Response · Authors · 2021-08-10
**General Response to all the Reviews**

We are grateful to all the reviewers for their time and effort spent in reviewing our paper, and thankful for the many thoughtful comments and suggestions. Three reviewers suggested accepting the paper (grade 7) and we are grateful for that. One reviewer has some concerns (grade 5) but we believe that we can address her/his concerns, see our answer below. We have addressed all the issues proposed by the four reviewers and will incorporate your suggestions in the final version of our paper.

---

### Decision · Program_Chairs · 2021-09-27

**Decision:**

Accept (Poster)

**Comment:**

This paper analyzes the step decay schedule (constant and then cut) for nonconvex optimization problems, showing that it can find an approximate first order stationary point in O(ln T/\sqrt{T}) rate. Most reviewers found the result interesting and that it gives a better understanding of the step decay schedule. There are some concerns that should be addressed in the revision: 1. clarifying why the requirement on boundedness of f can be replaced by E[f(x_t) - f(x^*)] and why the latter expectation can be bounded in natural cases; 2. detailed comparisons with previous results acknowledging that similar (or even better) rates were achieved by different algorithms in all the settings.